# Doc2B acts as a calcium sensor for vesicle priming requiring synaptotagmin-1, Munc13-2 and SNAREs

Sébastien Houy[1], Alexander J Groffen[2], Iwona Ziomkiewicz[1,3], Matthijs Verhage[2,4], Paulo S Pinheiro[1†], Jakob Balslev Sørensen[1*]

[1]Neuronal Secretion Group, Department of Neuroscience, University of Copenhagen, København, Denmark; [2]Department of Clinical Genetics, Center for Neurogenomics and Cognitive Research, VU Medical Center, Amsterdam, Netherlands; [3]Discovery Sciences, Innovative Medicines and Early Development, AstraZeneca R&D, Cambridge, United Kingdom; [4]Department of Functional Genomics, Faculty of Science, Center for Neurogenomics and Cognitive Research, VrijeUniversiteit, Amsterdam, Netherlands

*For correspondence:
jakobbs@sund.ku.dk

Present address: [†]Center for Neuroscience and Cell Biology, University of Coimbra, Coimbra, Portugal

**Abstract** Doc2B is a cytosolic protein with binding sites for Munc13 and Tctex-1 (dynein light chain), and two C2-domains that bind to phospholipids, $Ca^{2+}$ and SNAREs. Whether Doc2B functions as a calcium sensor akin to synaptotagmins, or in other calcium-independent or calcium-dependent capacities is debated. We here show by mutation and overexpression that Doc2B plays distinct roles in two sequential priming steps in mouse adrenal chromaffin cells. Mutating $Ca^{2+}$-coordinating aspartates in the C2A-domain localizes Doc2B permanently at the plasma membrane, and renders an upstream priming step $Ca^{2+}$-independent, whereas a separate function in downstream priming depends on SNARE-binding, $Ca^{2+}$-binding to the C2B-domain of Doc2B, interaction with ubMunc13-2 and the presence of synaptotagmin-1. Another function of Doc2B – inhibition of release during sustained calcium elevations – depends on an overlapping protein domain (the MID-domain), but is separate from its $Ca^{2+}$-dependent priming function. We conclude that Doc2B acts as a vesicle priming protein.

DOI: https://doi.org/10.7554/eLife.27000.001

## Introduction

The calcium signal triggering fast synchronous secretion in chemical synapses and neuroendocrine cells is confined to nanodomains or microdomains around calcium channels (*Neher, 2015*); it is of high amplitude, but brief duration, and calcium sensors that trigger synchronous release therefore display fast kinetics, but low affinity (*Pinheiro et al., 2016*). These sensors include synaptotagmin (syt)-1,-2 and -9 (*Xu et al., 2007*). Another class of calcium sensors has high affinity and slow kinetics; they are suited to detect increases in the spatially averaged calcium concentration following stimulation. These sensors – including Munc13 proteins – act in preparing vesicles for later release, that is, they promote vesicle priming by stimulating SNARE-complex formation (*Lipstein et al., 2012*; *Ma et al., 2013*; *Man et al., 2015*). For some calcium sensors, it remains controversial whether they belong to one or the other class – or to both.

A high-affinity calcium sensor in the adrenal medulla, pancreatic beta cells and central neurons is Doc2B (*Friedrich et al., 2008*; *Groffen et al., 2010*; *Ramalingam et al., 2012*). Doc2 proteins are named after their double C2-domains; they lack a transmembrane domain and calcium-binding therefore leads to the protein cycling on and off the plasma membrane, dictated by the electrical activity of the cell (*Groffen et al., 2006*). For Doc2B, half-maximal membrane binding occurs at ~0.2

µM calcium (*Groffen et al., 2006*). In neurons, Doc2B stimulates spontaneous release (*Groffen et al., 2010*; *Pang et al., 2011*). This function aligns well with the ability of Doc2B to bind to SNAREs, stimulate membrane stalk formation (*Brouwer et al., 2015*) and SNARE-dependent membrane fusion in vitro (*Groffen et al., 2010*) and would be consistent with a function of Doc2B in triggering fusion, at lower calcium concentrations than syt-1,-2 and -9. This was further supported by the finding that a permanently membrane-bound Doc2B, the DN-mutant, resulted in a high and calcium-independent mini release rate (*Groffen et al., 2010*). However, mutation of all six $Ca^{2+}$-coordinating aspartates created a mutant (denoted '6A'), which could not bind to $Ca^{2+}$, but still supported spontaneous release (*Pang et al., 2011*). It was instead suggested that Doc2B might perform an upstream $Ca^{2+}$-independent function in increasing spontaneous release (*Pang et al., 2011*). One group has reported that Doc2A acts as a calcium sensor involved in asynchronous release, which can also be supported by Doc2B following overexpression (*Yao et al., 2011*; *Gaffaney et al., 2014*; *Xue et al., 2015*). However, other groups reported no change in asynchronous release following elimination of Doc2B (*Groffen et al., 2010*), or all Doc2-proteins (*Pang et al., 2011*). Yet another proposed function of Doc2B is to serve as a scaffold by recruiting other factors to the release machinery. Munc13-proteins bind to the MID-domain in the N-terminal tail of the protein (*Orita et al., 1997*; *Mochida et al., 1998*), providing a mechanism to target Munc13-1 to the plasma membrane (*Friedrich et al., 2013*). In insulin releasing β-cells, secretion was impaired in the absence of Doc2B (*Ramalingam et al., 2012*), and it was suggested that Doc2B might serve as a scaffold for Munc18-1 and Munc18c (*Ramalingam et al., 2014*).

Data obtained after knockout of Doc2B in mouse adrenal chromaffin cells showed a dual effect: the size of the Readily Releasable Pool (RRP) of vesicles was decreased – especially during repetitive stimulation – and the sustained component of release was potentiated (*Pinheiro et al., 2013*). Consistently, upon overexpression of Doc2B, the RRP size increased and the sustained component was inhibited. Overall, Doc2B synchronizes release with the onset of the calcium signal. Here, we addressed the question whether those roles of Doc2B represent calcium dependent or independent functions. We further investigated whether the two functions of Doc2B in chromaffin cells require interaction with other proteins. We conclude that Doc2B plays distinct $Ca^{2+}$-independent and $Ca^{2+}$-dependent roles in two sequential vesicle priming steps, which support the Slowly Releasable Pool (SRP) and the RRP, respectively (*Bittner and Holz, 1992*; *von Rüden and Neher, 1993*; *Voets, 2000*). The effect of Doc2B to move vesicles into the RRP requires interactions with $Ca^{2+}$, ubMunc13-2 and SNAREs – and the presence of syt-1.

## Results

### Doc2B requires synaptotagmin-1 and Munc13-2 to stimulate priming

To demonstrate the role of Doc2B in chromaffin cells we first compared secretion from Doc2B knockout (KO) adrenal chromaffin cells with Doc2B KO cells overexpressing Doc2B (*Figure 1A*). Doc2B was expressed from a bicistronic Semliki Forest Virus construct co-expressing EGFP as an expression marker. Doc2A is not expressed in adrenal chromaffin cells (*Friedrich et al., 2008*; *Pinheiro et al., 2013*). Secretion was elicited using calcium uncaging and monitored using capacitance measurements. Vesicle priming in chromaffin cells is calcium dependent, such that a higher $[Ca^{2+}]_i$ before stimulation increases the size of the exocytotic burst (*Voets, 2000*). In the first set of experiments, we stimulated cells from a high basal $[Ca^{2+}]_i$ to maximize priming (*Figure 1A*, *insert in top panel*; 828.8 ± 24.3 nM for Doc2B KO cells, 935.7 ± 30.3 nM for KO cells overexpressing Doc2B WT). The results (*Figure 1A* mid panel shows the mean capacitance traces of all recorded cells) showed that expression of Doc2B had a dual effect: it increased the burst of secretion (defined as capacitance increase within 0.5 s of uncaging, *Figure 1A*), while it decreased sustained release (capacitance increase between 0.5 and 5 s after uncaging, *Figure 1A*). Because sustained release dominates at later times, total release over the course of the experiment (5 s) was reduced by Doc2B expression. Detection of liberated catecholamines was performed by amperometry, in parallel with capacitance measurements from the same cells (*Figure 1A*, bottom panel shows mean traces of all recorded cells). Following integration, the amperometric signal mimicked the capacitance trace, confirming those measurements and showing that the vesicles released catecholamines. However, amperometry cannot distinguish between fast and slow burst components due to the

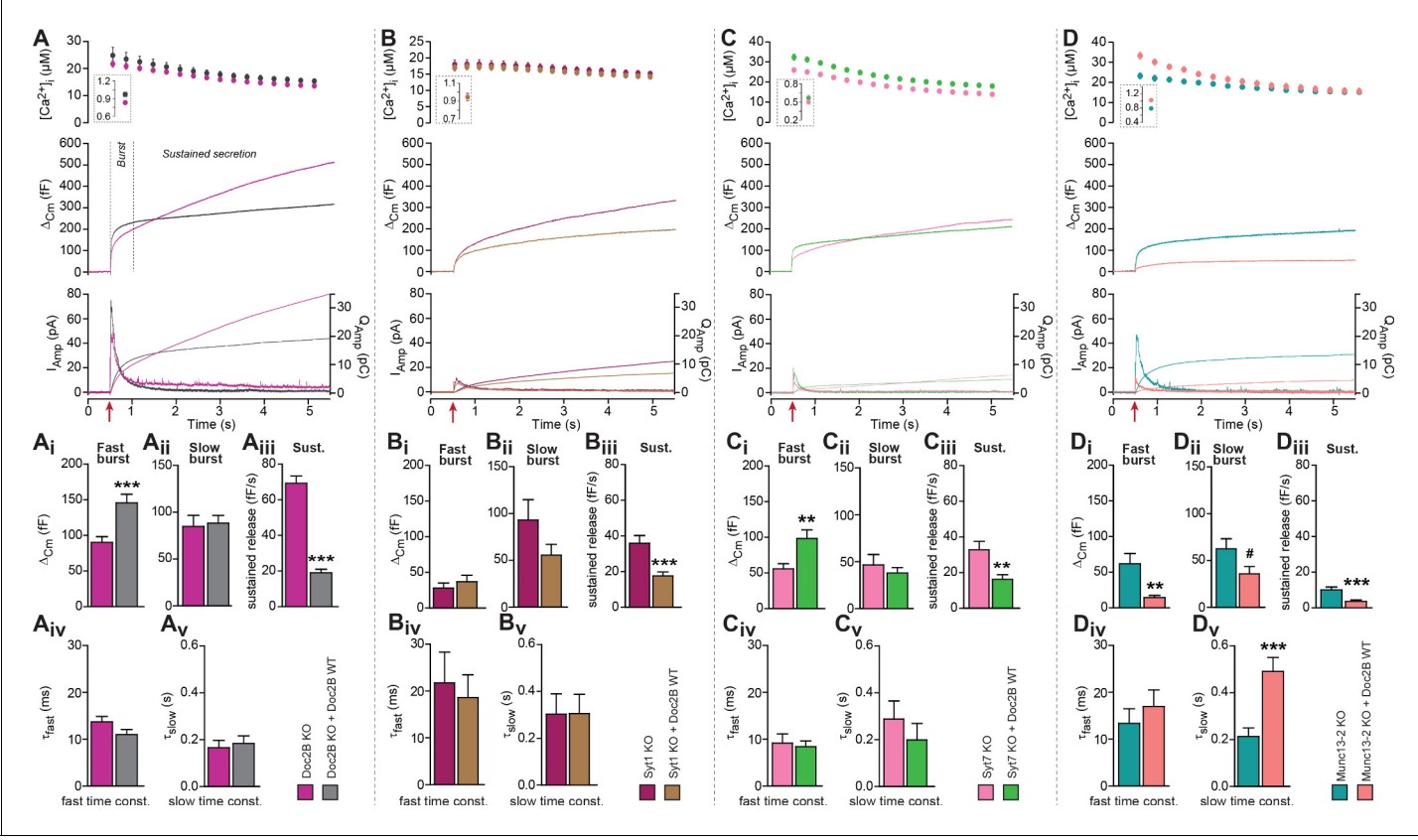

**Figure 1.** Doc2B requires synaptotagmin-1 and Munc13-2 to increase the Readily Releasable Pool (RRP) size. (A) Expression of Doc2B in Doc2B knockout (KO) mouse adrenal chromaffin cells. Top panels: Intracellular [$Ca^{2+}$] (mean ±SEM) after and (*insert*) shortly before $Ca^{2+}$ uncaging (uncaging light for 1–2 ms at red arrow). Middle panel: average capacitance traces from $Ca^{2+}$ uncaging experiments in Doc2B KO cells (magenta traces, n = 40 cells) and after overexpression of Doc2B WT in Doc2B KO cells (grey traces, n = 36 cells). Bottom panel: amperometric measurements (mean traces) from the same cells. Traces showing a release transient after the uncaging stimulus are the amperometric currents (left ordinate axes). Monotonically increasing traces display the time-integral of the amperometric current (i.e. the charge), scaled to the right ordinate axis. Note that the time-integrals display the same general shape as the capacitance traces, as expected. (Ai) The amplitude (mean ±SEM) of the fast burst was increased by Doc2B (***p<0.001, Mann-Whitney test), whereas (Aii) the amplitude of the slow burst component remained constant. Fast and slow burst correspond to the sizes of the RRP and the SRP, respectively. (Aiii) The rate of the near-linear sustained component (D) was strongly reduced by Doc2B (***p<0.001, Mann-Whitney test). (Aiv) The fusion time constant for the fast burst, and (Av) fusion time constant for the slow burst were unaffected by Doc2B. (B) $Ca^{2+}$ uncaging experiment in syt-1 KO cells (n = 18) and syt-1 KO cells overexpressing Doc2B WT (n = 20). Panels (B–Bv) are arranged as in A. Doc2B expression led to a highly significant reduction in sustained release, but no significant changes in the fast or slow burst sizes. ***p<0.001, Mann-Whitney test. (C) $Ca^{2+}$ uncaging experiment in syt-7 KO cells (n = 17) and syt-7 KO cells overexpressing Doc2B WT (n = 17). Panels (C–Cv) are arranged as in A. Doc2B expression led to a significant increase in fast burst size (**p<0.01, Mann Whitney test), and a reduction in sustained release (**p<0.01, unpaired, two-tailed Student's *t* test), as in control experiments (panel A). (D) $Ca^{2+}$ uncaging experiment in Munc13-2 KO cells (n = 16) and Munc13-2 KO cells overexpressing Doc2B WT (n = 18). Panels (D–Dv) are arranged as in A. All components of release were reduced by overexpression of Doc2B WT in the absence of Munc13-2 (**p<0.01, ***p<0.001, Mann Whitney test; # p=0.0575, unpaired two-tailed Student's t-test). The time constant of the slow component was increased (***p<0.001, unpaired two-tailed Student's t-test).

DOI: https://doi.org/10.7554/eLife.27000.002

The following figure supplements are available for figure 1:

**Figure supplement 1.** Overexpression of Doc2B in wild-type cells increases the burst size and decreases the sustained release.
DOI: https://doi.org/10.7554/eLife.27000.003

**Figure supplement 2.** No difference of expression of Doc2B in Syt-1 and Syt-7 WT vs KO mouse adrenal glands.
DOI: https://doi.org/10.7554/eLife.27000.004

diffusional delay. As a control, we also expressed Doc2B in WT chromaffin cells (CD1 mice and – in separate experiments – Black6 mice), which express endogeneous Doc2B, and we found similar changes: an increase in burst release, and a decrease in the sustained component (*Figure 1—figure supplement 1A,B*). We conclude that overexpression of Doc2B to boost endogenous levels can be

used to assay these two opposing effects of Doc2B in adrenal chromaffin cells. Previous data comparing WT to Doc2B KO cells identified consistent changes: a decrease in burst release (seen most clearly during depolarization protocols), and an increase in sustained release upon elimination of Doc2B (*Pinheiro et al., 2013*). Thus, even though overexpression using Semliki vectors boost protein levels far beyond endogenous levels, the effects on secretion are consistent with those seen in the presence of endogenous expression levels.

We performed kinetic analysis to quantify the changes induced by Doc2B expression in fast and slow burst secretion, which originate from the Readily Releasable vesicle Pool (RRP) and the Slowly Releasable Pool (SRP), respectively (*Voets, 2000*; *Walter et al., 2013*). These analyses showed that the fast burst amplitude – corresponding to RRP size – was increased by Doc2B expression (*Figure 1Ai*, *Figure 1—figure supplement 1Ai,Bi*), whereas slow burst amplitude – corresponding to SRP size – was unchanged when expressing Doc2B in Doc2B KO cells (*Figure 1Aii*), or after expression of Doc2B in Black 6 WT cells (*Figure 1—figure supplement 1Bii*), although it was increased when expressing Doc2B in CD1 WT cells (*Figure 1—figure supplement 1Aii*). The sustained component was strongly reduced in all three mouse lines (*Figure 1Aiii*, *Figure 1—figure supplement 1Aiii,Biii*). The time constants of fusion of fast and slow burst fusion were not significantly changed (*Figure 1Aiv,Av*, *Figure 1—figure supplement 1Aiv, Av, Biv, Bv*). Thus, Doc2B has opposing effects on RRP size and sustained release, whereas the effect on the SRP differs between the two mouse lines (Black6 and CD1). Note that our KO mice are backcrossed to a Black6 background.

A promoting effect on RRP size implies that Doc2B stimulates RRP filling (priming) during period of rest (i.e. before stimulation). We next wanted to understand whether Doc2B requires the presence of other exocytotic priming proteins in the cell in order to stimulate priming, or whether it acts independently. Synaptotagmin-1 (syt-1) knockout (KO) chromaffin cells display a smaller-than-normal fast burst (RRP), whereas the slow burst is preserved (*Voets et al., 2001*; *Mohrmann et al., 2013*); this slow burst depends on synaptotagmin-7 (syt-7) (*Schonn et al., 2008*). Overexpression of syt-1 on the wildtype background can increase the size of the RRP further (*Nagy et al., 2006*), indicating that syt-1 abundance controls the step converting SRP to RRP. Both Syt-1 KO and Syt-7 KO express endogenous Doc2B (*Figure 1—figure supplement 2*). We overexpressed Doc2B in syt-1 KO cells, and found that it did not increase the fast bursts in these cells, but it still led to a significant decrease in the sustained component (*Figure 1B,Bi–iii*). There was a non-significant decrease in slow burst in overexpressing cells, which might have been influenced by the reduction of the sustained component, which during kinetic analysis might 'leak into' the slow burst component. Changes in fusion time constants were not detected (*Figure 1Biv,Bv*). Thus, Doc2B acts upstream of or in cooperation with syt-1 to increase the RRP size, not in a redundant pathway. In syt-7 KO chromaffin cells all the components of release were depressed compared to WT cells (*Schonn et al., 2008*) (*Figure 1C*). In syt-7 KO chromaffin cells, expression of Doc2B caused an increase of the fast burst and a decrease of the sustained component (*Figure 1C,Ci–Cv*), indicating that Doc2B does not rely on syt-7. Amperometric measurements delivered independent evidence for the increase in burst release and decrease in sustained component upon overexpression of Doc2B in the syt-7 KO.

Munc13-proteins are the canonical priming proteins in neurons (*Augustin et al., 1999*) and chromaffin cells (*Ashery et al., 2000*), acting by opening syntaxin-1 within the confines of Munc18 (*Ma et al., 2013*) to facilitate SNARE-complex assembly. ubMunc13-2 is the dominating Munc13 isoform in adrenal chromaffin cells, and in its absence the fast burst of release was reduced due to a priming defect (*Man et al., 2015*). Interestingly, expressing Doc2B WT in ubMunc13-2 KO cells not only failed to support the increase in fast burst amplitude, but led to near-abolishment of all secretory phases (*Figure 1D,Di–Dv*), similar to the defect seen when eliminating SNARE-proteins (*Sørensen et al., 2003b*; *Borisovska et al., 2005*). This effect was confirmed by amperometry measurements (*Figure 1D*). Thus, Munc13-2 is required for the promoting functions of Doc2B on priming, and negative functions of Doc2B dominate in its absence.

## Mutated Doc2B renders vesicle priming Ca²⁺-independent

Vesicle priming in chromaffin cells is $Ca^{2+}$-dependent, with higher $Ca^{2+}$ concentrations in the sub-μ M range leading to a larger burst of release (*Bittner and Holz, 1992*; *von Rüden and Neher, 1993*; *Voets, 2000*). This mechanism is physiologically important, because it makes it possible for G-protein coupled receptors, which stimulate $Ca^{2+}$-release from intracellular stores to increase $[Ca^{2+}]$ in

the sub-µM range, to act synergistically with neuronal stimulation to strongly increase adrenaline output (*Teschemacher and Seward, 2000*). Since Doc2B is a priming protein with $Ca^{2+}$-dependent trafficking, we next investigated the possibility that it might act as a $Ca^{2+}$-sensor for vesicle priming. To this end, we mutated Doc2B's $Ca^{2+}$-binding sites. Mutating two of the coordinating aspartate residues in the C2A-domain to asparagine (D218N/D220N, denoted DN-mutation) causes Doc2B to reside permanently at the plasma membrane (*Groffen et al., 2006*; *Friedrich et al., 2008*; *Gaffaney et al., 2014*; *Xue et al., 2015*; *Michaeli et al., 2017*). Immunostainings against Doc2B and synaptobrevin-2(syb-2)/VAMP2 are shown in *Figure 2A and B*. The DN-mutant was expressed at similar levels as WT Doc2B (*Figure 2A,C*), and the vesicular marker VAMP2/synaptobrevin-2 staining also showed similar levels in separate stainings (*Figure 2B,D*). Inspecting the stainings, both the WT and the DN mutant appeared to be at or close to the plasma membrane, but chemical fixation can cause $Ca^{2+}$-influx, which induces trafficking of Doc2B at the sub-µM level; thus, membrane localization cannot be assess from staining of fixed samples. Instead, we used an EGFP fusion protein and live cell confocal imaging to assess localization (*Figure 3*). These measurements verified previous findings and showed that Doc2B readily traffics between cytosol and plasma membrane during reversible exposure to a High-$K^+$ solution (*Figure 3A–C*)(*Groffen et al., 2006*). In contrast, the DN-mutant was present at the membrane at higher levels than the WT protein at rest (*Figure 3A,B*). Nevertheless, additional trafficking of the DN-mutant to the membrane was seen during High-$K^+$ exposure, due to $Ca^{2+}$-binding to the C2B-domain (*Figure 3C*).

For physiological measurements, we expressed the DN-mutant and EGFP separately from a bicistronic message (*Figure 4*). To specifically test $Ca^{2+}$-dependent priming, we adjusted the $Ca^{2+}$ concentration of our pipette solution to either ~200 nM (*Figure 4A*), or to ~900 nM (*Figure 4G*). Final adjustments (increases) in the $[Ca^{2+}]$ were achieved using monochromator light (which was used simultaneously to monitor $[Ca^{+2}]$) to uncage some of the $Ca^{2+}$-cage ~20 s before stronger stimulation using a flash UV-lamp to probe secretion itself (*Voets, 2000*). When we carried out $Ca^{2+}$ uncaging experiments from a low resting $[Ca^{2+}]_i$ (~200 nM, *Figure 4A*) expression of Doc2B WT caused an increase in slow burst release and a decrease in the sustained release rate (*Figure 4A–D*). Strikingly, when expressing the Doc2B DN-mutation, the burst was potentiated even more (orange curve, *Figure 4A*). The potentiation was seen both in the fast and the slow burst components (*Figure 4B, C*). In experiments with a high basal $[Ca^{2+}]_i$ (*Figure 4G*), both Doc2B KO cells and KO cells overexpressing Doc2B responded with a larger exocytotic burst (compare secretion within the first 0.5 s after the stimulation in *Figure 4A and G*), showing that calcium-dependent priming was successfully induced. However, under these conditions, the responses of DN-mutant cells were now indistinguishable from the WT, for all secretory phases (*Figure 4G–J*). At our high basal $[Ca^{2+}]_i$ condition (~900 nM), Doc2B WT will be fully $Ca^{2+}$-saturated and membrane localized (*Groffen et al., 2006*). Strikingly, when comparing the burst amplitude (fast + slow burst) in the DN-mutation with the WT, it is clear that the DN-mutation rendered burst amplitude independent of preflash-$[Ca^{2+}]_i$ (*Figure 4M*). This indicates that $Ca^{2+}$ binding to WT Doc2B drives priming of new vesicles, or that this reaction is bypassed when Doc2B is permanently localized at the plasma membrane (see Figure 10 and Discussion). Therefore, the DN-mutant is a gain-of-function mutant in chromaffin cells when studied from a low basal $[Ca^{2+}]$. Kinetic analysis indicated that the DN-mutant decreased the time constant for fast fusion (*Figure 4K*), perhaps indicating a separate property of this protein. The inhibitory effect was unchanged by the DN-mutation, whether probed from high or low basal $[Ca^{2+}]_i$ (*Figure 4D,J*). Thus, Doc2B has separate positive and negative effects on secretion.

The 6A-mutation, which includes aspartate-to-alanine mutations of six key $Ca^{2+}$-binding aspartate residues, three in each C2-domain (C2A: D163A/D218A/D220A, C2B: D303A/D357A/D359A) blocks $Ca^{2+}$ binding to both C2-domains (*Pang et al., 2011*). Although the 6A-mutant cannot bind to $Ca^{2+}$, it was able to rescue the normal mEPSC frequency in cultured neurons (*Pang et al., 2011*). This finding was used to argue that Doc2B has calcium-independent upstream effects on miniature release. Here, using live cell imaging of a Doc2B(6A)-EGFP fusion protein we showed that the 6A-mutant localizes permanently to the plasma membrane in excess of the WT protein, and it does not traffic in response to depolarization (*Figure 3A,B,C*). Similar results were previously reported using a quadruple aspartate-to-asparagine mutation in C2A/B (*Gaffaney et al., 2014*).

When expressed from a bicistronic message, the 6A-mutant reached protein levels similar to those of WT Doc2B (*Figure 2A–D*). From low basal $[Ca^{2+}]_i$, this mutation caused a significant increase in the slow and fast burst of release (*Figure 5A,B*). However, from high basal $[Ca^{2+}]_i$, the

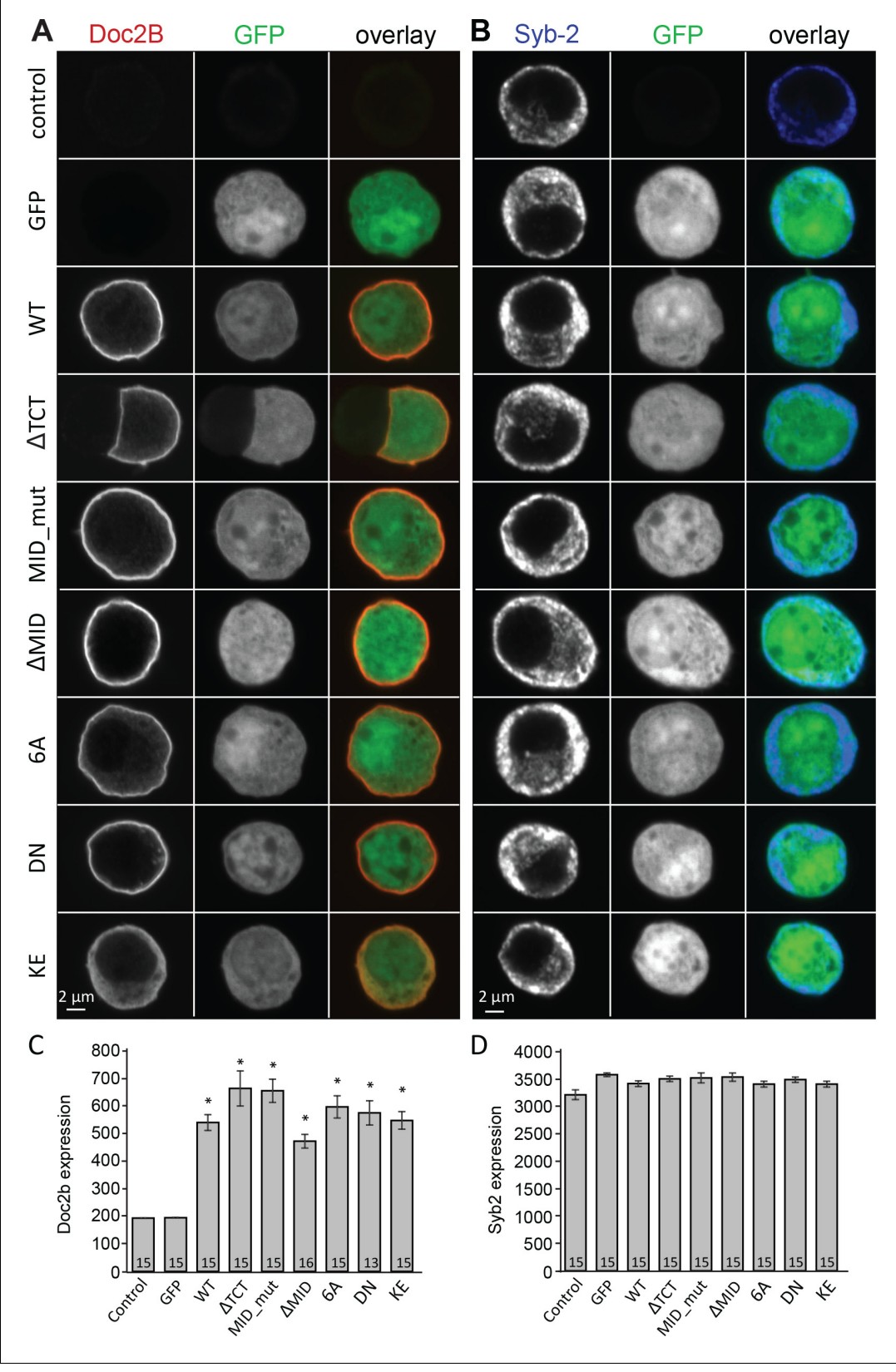

**Figure 2.** Overexpression of Doc2B in Doc2A/B deficient chromaffin cells transfected with semliki vectors. (A–B) Typical chromaffin cells stained for (A) Doc2B or (B) VAMP2/synaptobrevin-2 to mark chromaffin granules. EGFP was co-expressed from an internal ribosome entry site (IRES) to mark semliki infection. 'Control' indicates non-
*Figure 2 continued on next page*

*Figure 2 continued*

infected cells. WT: wildtype Doc2B; ΔTCT: deletion of aa. 2–9 to block Tctex-1 binding; MID_mut: substitution of aa. 15–20 (QEHMAI to YKDWAF) to block Munc13 interaction; ΔMID: deletion of aa. 14–41 to block Munc13 interaction; 6A: D163,218,220,303,357,359A; DN: D218,220N, KE: K237,319E. (**C–D**) Average staining intensity ±sem for (**C**) Doc2B and (**D**) VAMP2/synaptobrevin-2 from the indicated number of cells. *p<0.001 (t-test for independent samples with Bonferroni correction).

DOI: https://doi.org/10.7554/eLife.27000.005

fast burst of release remained unchanged compared to the Doc2B KO cells (*Figure 5G,H*). When plotting the burst size as a function of basal $[Ca^{2+}]_i$, cells expressing the 6A-mutation displayed a calcium-independent burst size, intermediate between the bursts supported by Doc2B WT at low and high basal $[Ca^{2+}]_i$ (*Figure 5M*). The invariance of the burst size with respect to basal $[Ca^{2+}]_i$ again shows that membrane localization of Doc2B occludes $Ca^{2+}$-dependent priming. However, the 6A-mutant is not quite as efficient as the fully $Ca^{2+}$-loaded WT Doc2b-protein – or the DN-mutant – since the maximal level of priming that it supports is smaller than after WT rescue, which is due to the inability to increase RRP size at high basal $[Ca^{2+}]_i$.

Half-maximal trafficking of Doc2B to the plasma membrane requires ~200 nM $Ca^{2+}$ (*Groffen et al., 2006*), which coincides with resting $[Ca^{2+}]$ in the experiments in *Figure 3A and B*. Due to the finite affinity of our calcium cage, nitrophenyl-EGTA, which has a $K_d$ of 80 nM (*Ellis-Davies and Kaplan, 1994*) it was not feasible to conduct calcium uncaging experiments from even lower basal $[Ca^{2+}]_i$. This would cause $Ca^{2+}$ to dissociate from the cage before uncaging, leaving the unbound cage as an additional $Ca^{2+}$-buffer that would prevent increases in $[Ca^{2+}]$. Instead, we probed the DN-mutant using a depolarization protocol and a pipette solution including 0.5 mM EGTA without added $Ca^{2+}$ (*Figure 6*). It is commonly assumed that six brief (10 ms) depolarizations elicit the fusion of the Immediately Releasable Pool (IRP), which are RRP-vesicles co-localized with $Ca^{2+}$-channels, whereas four subsequent longer (100 ms) depolarizations fuse the entire RRP (*Voets et al., 1999*). We compared Doc2B KO cells expressing Doc2B WT and Doc2B DN. The depolarization protocol caused a moderate increase in $[Ca^{2+}]_i$, to ~2 µM in cells expressing either construct (*Figure 6A*). The modest increase in $[Ca^{2+}]_i$ is due to the strong inhibition of $Ca^{2+}$-channels induced by Doc2B overexpression in chromaffin cells (*Toft-Bertelsen et al., 2016*). Therefore, these data cannot be directly compared to findings in Doc2B KO or WT cells (*Pinheiro et al., 2013*). The DN-mutant did not induce any significant increase in IRP size over Doc2B WT (*Figure 6A–B*), but the capacitance after four long depolarizations was significantly increased (*Figure 6A,C*; we denote the increase 'RRP' for the sole reason that it was comparable to the fast burst measured by $Ca^{2+}$ uncaging in *Figure 4A–B*). When repeating the protocol (6 + 4 depolarizations), the effect of the DN mutation on 'RRP' size was largest in the first two trials and subsided thereafter as the cells underwent secretory run-down (*Figure 6D*). These experiments show that the DN-mutation specifically increased the RRP size, but not the IRP.

Overall, we conclude from these experiments that Doc2B is a calcium-dependent priming protein acting to increase the size of the secretory burst (consisting of the RRP and the SRP vesicles), and calcium-dependent priming can be occluded by mutations at the $Ca^{2+}$- and phospholipid-binding interface within its C2 domains. However, the decrease of the sustained component is unchanged by mutating the $Ca^{2+}$ binding sites of Doc2B and must, therefore, represent a distinct function of Doc2B.

## SNARE interactions are necessary for the stimulation of priming by Doc2B

The poly-lysine stretch on the C2B-domain of syt-1 mediates the interaction with SNAREs and PI(4,5)P$_2$ (*Bai et al., 2004*; *Rickman et al., 2006*), and the positive charge is important for triggering release (*Li et al., 2006*). The Doc2B protein contains a similar polylysine stretch at its C2B domain, and also a smaller stretch of basic residues on the C2A domain. Mutating a lysine to glutamate in each C2 domain creates the KE-mutant (K237E/K319E), which was previously shown to display a profound loss of binding to the SNARE-complex, while retaining $Ca^{2+}$-dependent binding to liposomes (*Groffen et al., 2010*). We recently found that Doc2B overexpression in adrenal chromaffin cells causes an increase in syntaxin-1 immunoavailability, due to the recruitment of syntaxin-1 from plasma

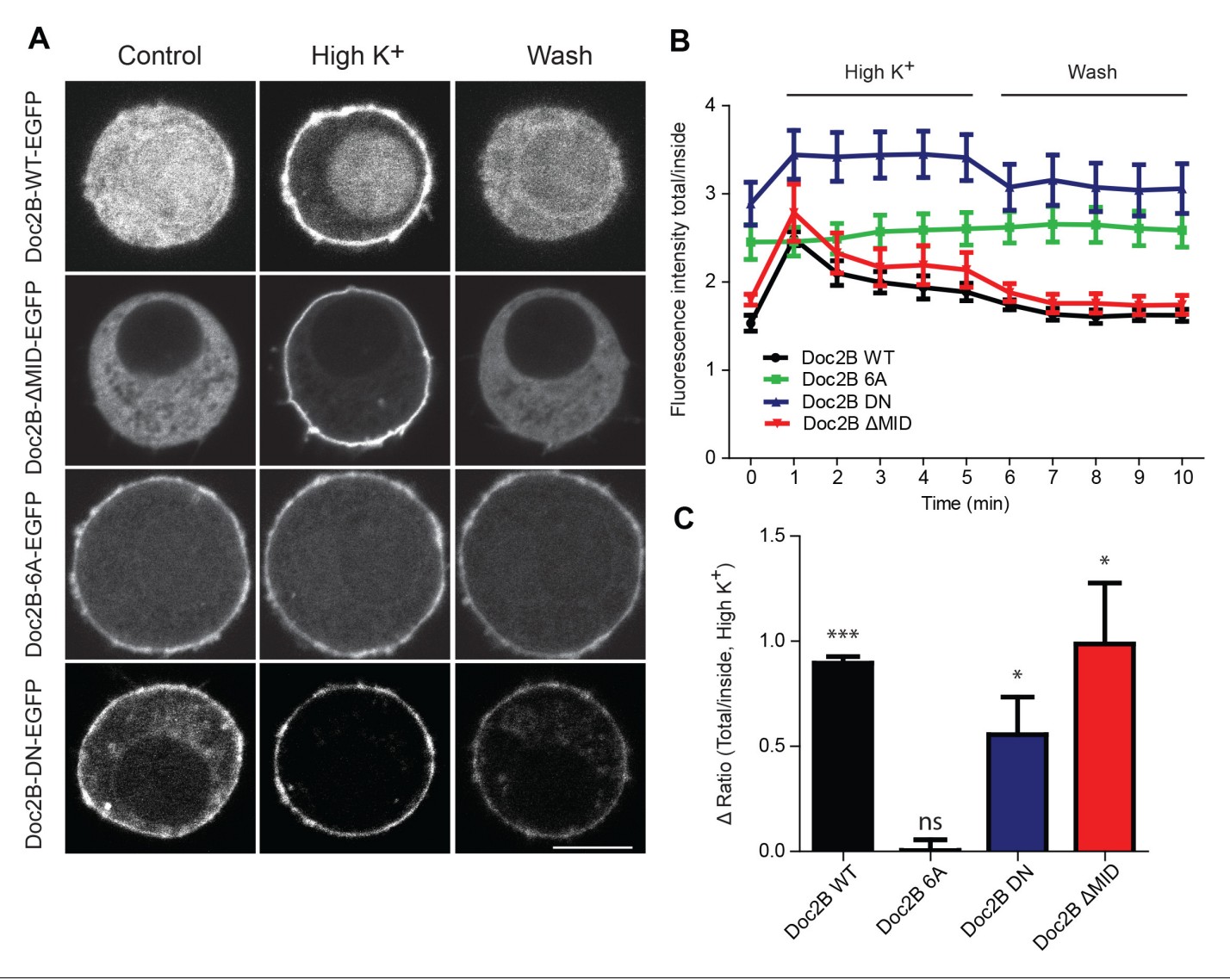

**Figure 3.** Doc2B translocates to the plasma membrane in a calcium-dependent manner. (**A**) Live confocal imaging experiments in CD1 mouse adrenal chromaffin cells expressing Doc2B WT, Doc2B ΔMID, Doc2B 6A or Doc2B DN fused to EGFP. Cells depolarization with High K$^+$ solution (59 mM) triggers the recruitment of Doc2B WT, Doc2b ΔMID and in a lesser extent Doc2B DN to the plasma membrane. Mutations of the calcium binding residues of both C2 domains of Doc2B (Doc2B 6A) led to a permanent localization at the plasma membrane. Washing with extracellular solution (Wash) permits Doc2B WT and ΔMID to re-localize in the cytosol. Scale bar = 5 μm (**B**) Quantification of Doc2B intensity (ratio total intensity/cytosol intensity) during live imaging experiment. Cells were imaged at resting and after 1 to 5 min stimulation (high K+) every minute, and then washed with extracellular solution and imaged every minute for 5 min. (**C**) Quantification of the plasma membrane recruitment of doc2B WT and mutants after 1 min stimulation with high K$^+$ solution (Difference in ratio [total fluorescence intensity/cytosol fluorescence intensity] at 1 min compared to resting) showing that only Doc2b 6A does not traffic (n cells: Doc2B WT: 4; Doc2B 6A: 5; Doc2b DN: 11; Doc2B ΔMID: 8. Panel C: One-sample t-test comparing the difference in fluorescence ratio to zero; * p-value<0,05; *** p-value<0.001).
DOI: https://doi.org/10.7554/eLife.27000.006

membrane clusters (*Toft-Bertelsen et al., 2016*). We therefore stained for syntaxin-1 after expression of Doc2B WT and Doc2B KE in adrenal chromaffin cells (*Figure 7A*). Using expression of Botulinum Toxin C, we previously demonstrated that this staining is specific for syntaxin-1 (*Toft-Bertelsen et al., 2016*). The results indeed showed that syntaxin-1 staining was more intense after expressing Doc2B WT or Doc2B DN (*Figure 7B*), whereas expression of Doc2B KE did not change syntaxin-1 staining intensities, consistent with the loss of SNARE binding by this mutation.

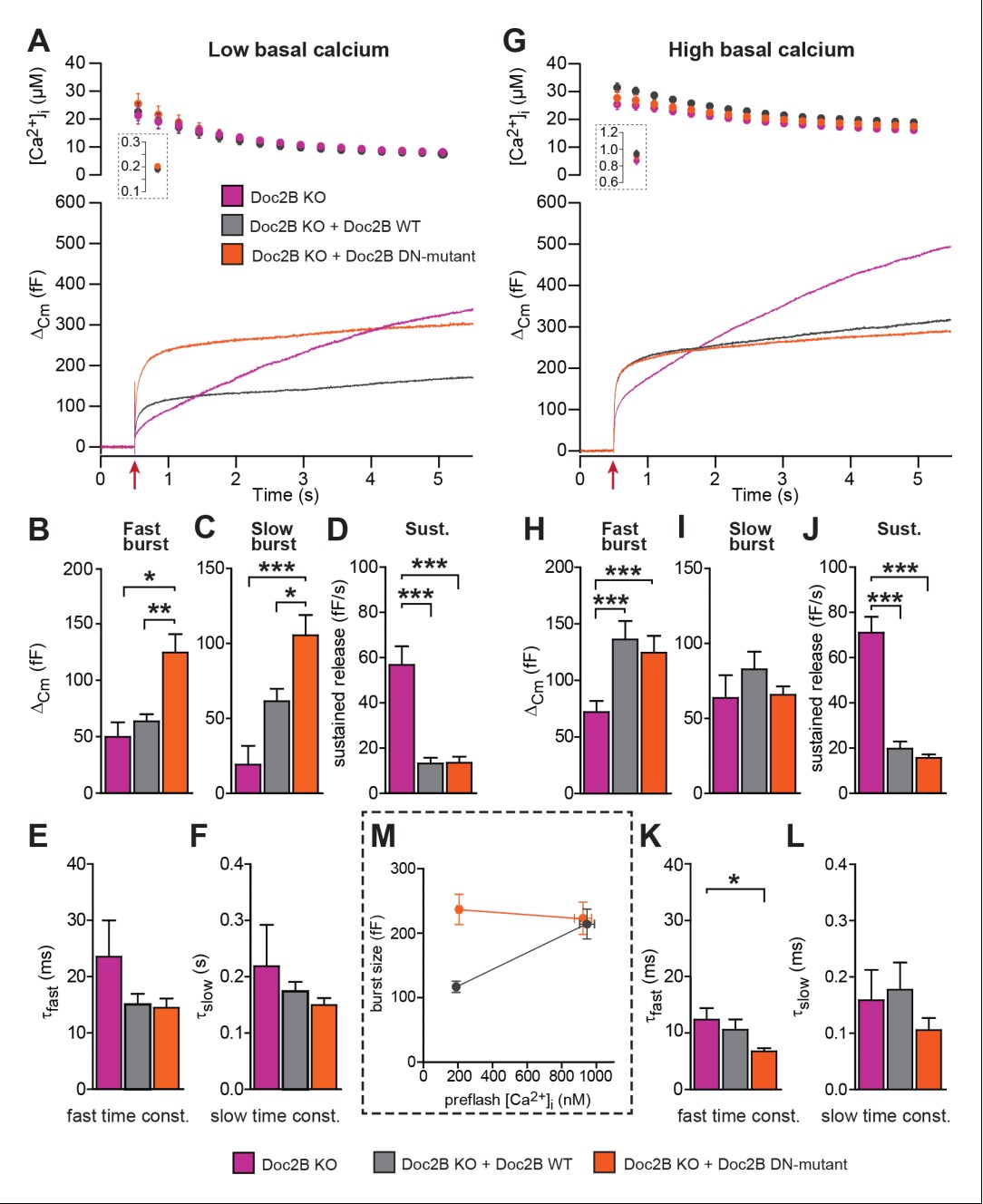

**Figure 4.** The Doc2B DN-mutant (D218,220N) supports maximal vesicle priming independently of the $Ca^{2+}$ level. (A-F) at low basal (preflash) calcium, the DN-mutant supports more secretion than WT Doc2B. (A) $Ca^{2+}$ uncaging experiments in Doc2B KO cells (n = 12), Doc2B KO cells overexpressing Doc2B WT (n = 19) and Doc2B KO cells overexpressing Doc2b DN (n = 20 cells). Top panel shows the average (±SEM) post-flash $[Ca^{2+}]_i$ measured by microfluorimetry; inset shows the pre-flash $[Ca^{2+}]_i$. Bottom panel shows the changes in cell membrane capacitance (mean traces for all cells). (B–F) Kinetic analysis revealed that both the fast and slow bursts were larger upon overexpression of Doc2B DN than of Doc2B WT, leading to a gain of function phenotype at low basal cell calcium. The sustained component was similarly decreased by overexpression of wildtype and DN-mutated Doc2B (*p<0.05, **p<0.01, ***p<0.001, Dunn's multiple comparison test). (G), similar experiment to (A) but in high basal pre-flash $Ca^{2+}$ (Doc2B KO, n = 20 cells; Doc2B KO expressing Doc2B WT, 20 cells; Doc2B KO expressing Doc2B DN, 32 cells). (H–L) Kinetic analysis shows that overexpression of either Doc2B DN or Doc2B WT causes a similar increase in the fast burst. Similar to the low preflash $[Ca^{2+}]_i$ experiment, the sustained component was decreased to the same levels by overexpression of both proteins. (K) Overexpression of Doc2B DN caused a significant

*Figure 4 continued on next page*

*Figure 4 continued*
acceleration of the fast burst fusion kinetics. (M) Plot of the total burst size (fast + slow) as a function of the basal cell calcium reveals that Doc2B DN supports maximal LDCV priming irrespective of the cell Ca$^{2+}$. *p<0.05, ***p<0.001; Dunn's multiple comparison test.
DOI: https://doi.org/10.7554/eLife.27000.007

Upon expression in Doc2B KO chromaffin cells (expression level in *Figure 2*), the Doc2B KE-mutant failed to support the increase in fast burst release seen upon expression of the WT protein (*Figure 7C–D*). However, the sustained component was depressed by the KE-mutant to similar levels as by the WT protein (*Figure 7F*), while the kinetics of fusion were unchanged (*Figure 7G–H*). Therefore, we conclude that SNARE interaction is necessary for the ability of Doc2B to increase the RRP. This agrees with previous data obtained for overexpressed Doc2A in PC12 cells (*Sato et al., 2010*). We cannot, however, rule out the alternative that Doc2B:PI(4,5)P$_2$ interactions could drive this function. In syt-1, lysines can interact alternatively with SNAREs or PI(4,5)P$_2$ (*Brewer et al., 2015*; *Park et al., 2015*; *Zhou et al., 2015*; *Schupp et al., 2016*), and recently it was found that K319 in Doc2B interacts with plasma membrane PI(4,5)P$_2$ (*Michaeli et al., 2017*).

The dynein light chain Tctex-1 was identified as a Doc2-interacting partner in a yeast two-hybrid screen using the N-terminal region of Doc2A as a bait (*Nagano et al., 1998*). Both Doc2A and Doc2B interact with tctex-1, and the interaction site maps to the first eight amino acids (*Orita et al., 1997*). In an attempt to identify the molecular background for the inhibitory function of Doc2B on vesicle priming during sustained Ca$^{2+}$ elevation, we performed rescue experiments with a Doc2B mutant, where amino acids 2–9 were deleted (ΔTCT). The ΔTCT-mutant was expressed similarly to WT Doc2B (*Figure 2*) and rescued the burst of secretion (secretion within 0.5 s) almost as well as the WT Doc2B (*Figure 8A*). No significant differences between the Doc2B ΔTCT and Doc2B WT were identified (*Figure 8A–C*). The ΔTCT mutant also still recapitulated the inhibitory function of Doc2B on the sustained component of release (*Figure 8D*), which, therefore, does not depend on binding to tctex-1.

## The MID-domain is essential for both positive and negative functions of Doc2B

Both Doc2A and Doc2B interact with Munc13-1 through the Munc13 Interacting Domain (MID), which is located near the N-terminal end of Doc2B, encompassing amino acids 14 to 38 (sequence in Doc2B: see *Figure 9A*)(*Orita et al., 1997*). Due to the MID domain Doc2B can recruit Munc13-1 to the membrane (*Friedrich et al., 2013*). Both the MID domain in Doc2B and the C1 domain of Munc13-1 are required for that trafficking event (*Friedrich et al., 2013*). Conversely, Munc13-1 might traffic to the plasma membrane by binding to phorbol esters through its C1 domain, bringing Doc2B along (*Duncan et al., 1999*); however, this has recently been questioned (*Friedrich et al., 2013*). Overall, the effects of Doc2B on priming might depend on interactions with Munc13 proteins, as already indicated by the lack of a positive priming effect when expressing Doc2B in Munc13-2 KO cells (*Figure 1D*).

To test for this possibility, we first used a mutation (MID_mut) of the MID-domain (*Figure 9A*), which was previously shown to eliminate the Doc2-Munc13 interaction in a cell-free pull-down assay, and to block synaptic transmission upon infusion into cholinergic synapses (*Mochida et al., 1998*). This mutant does not co-translocate to the plasma membrane after phorbol ester stimulation of Munc13, but still translocates after Ca$^{2+}$ stimulation of Doc2b (*Groffen et al., 2004*). Expression of the MID_mut in Doc2B KO chromaffin cells (expression level in *Figure 2*) showed that this mutation did not support an increase in the fast burst (RRP) of release, as seen in parallel experiments after expression of Doc2B WT (*Figure 9B*), indicating that the MID domain is necessary for this function. However, sustained secretion was still suppressed (*Figure 9Biii*), and the fusion kinetics were unaffected (*Figure 9Biv–Bv*). Therefore, the mutation shows – like the KE-mutation and the Ca$^{2+}$-ligand mutations – that the positive and negative functions of Doc2B can be separated by mutation.

The lack of an effect of the MID_mut on sustained release was puzzling, since so far it had been assumed that Munc13 binding is a prerequisite for the function of Doc2A/B in secretion (but see [*Gaffaney et al., 2014*]). Therefore, we decided to remove the entire MID domain, through the deletion of amino acids 14–41 (*Figure 9A*, Doc2B ΔMID). Importantly, this mutation was expressed at

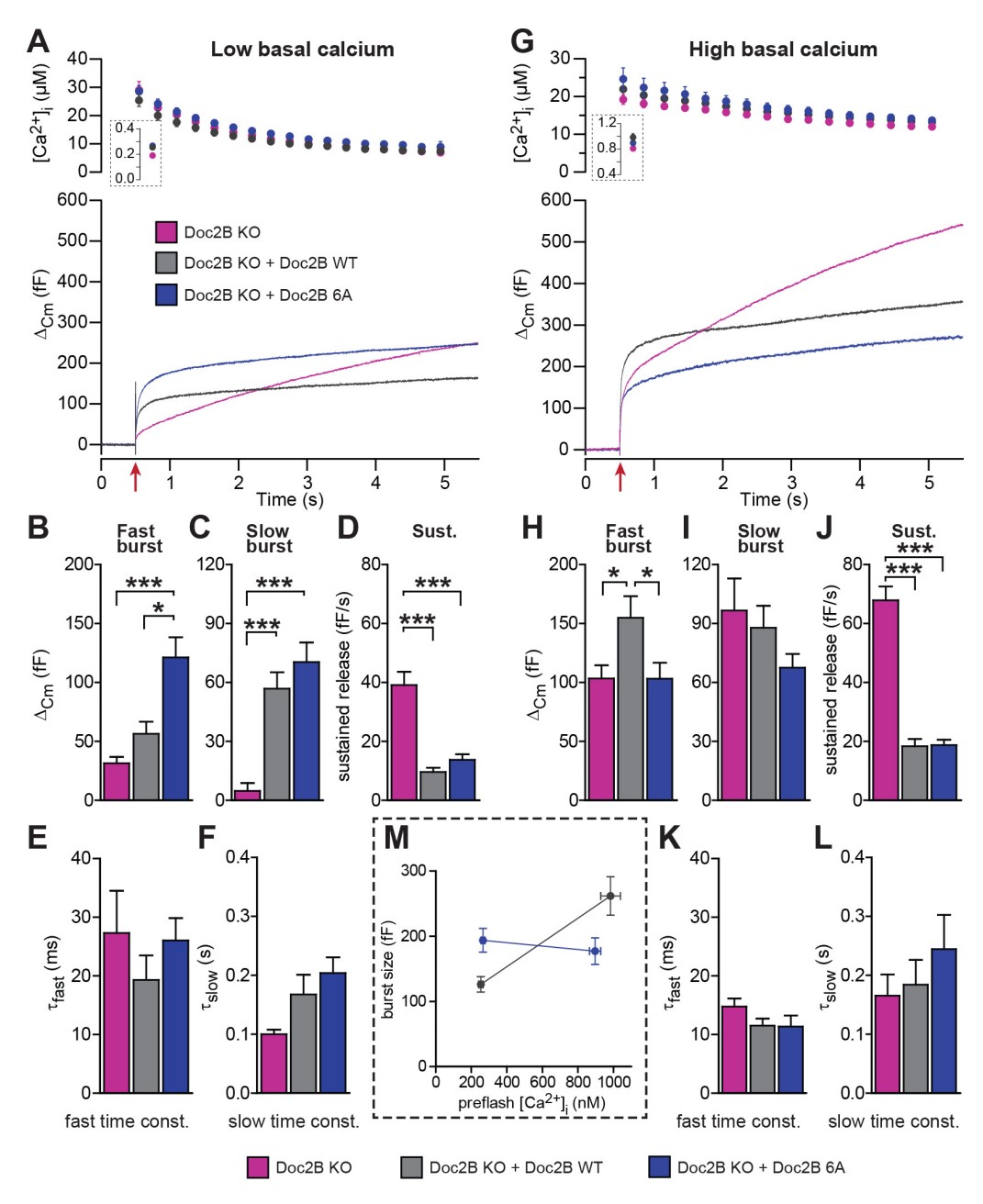

**Figure 5.** Mutation of the Ca²⁺binding sites of Doc2B leads to permanent membrane localization and Ca²⁺-independent priming. (A-F) When stimulated from low basal calcium, the Ca²⁺-binding mutant Doc2B 6A supports more secretion that Doc2B WT. (A) Ca²⁺ uncaging in Doc2b KO cells (n = 23), KO cells overexpressing Doc2B WT (n = 19) and KO cells overexpressing Doc2B 6A (n = 18). Panels are arranged as in *Figure 4*. (B–F) Kinetic analysis revealed that the fast burst was increased more by overexpression of Doc2B 6A than by Doc2B WT, showing a gain of function of the 6A mutant at low basal cell calcium. The sustained component was similarly decreased by overexpression of either protein. *p<0.05, ***p<0.001; Dunn's multiple comparison test. (G) similar experiment to (A) but stimulated from high basal pre-flash [Ca²⁺]ᵢ (Doc2B KO, n = 25 cells; Doc2B WT, n = 21 cells; Doc2B 6A, n = 22 cells). (H–L) Kinetic analysis shows that overexpression of Doc2B WT caused a marked increase in the fast burst while Doc2B 6A did not, demonstrating a loss of function when studied from a high basal [Ca²⁺]. Similar to the low [Ca²⁺]ᵢ experiment, the sustained component was decreased to the same levels by overexpression of both proteins. *p<0.05, Tukey's multiple comparison test; ***p<0.001, Dunn's multiple comparison test. (M) Plot of the total burst size (fast + slow) as a function of the basal cell calcium reveals the Ca²⁺-independent priming reaction in the presence of the 6A mutant.

DOI: https://doi.org/10.7554/eLife.27000.008

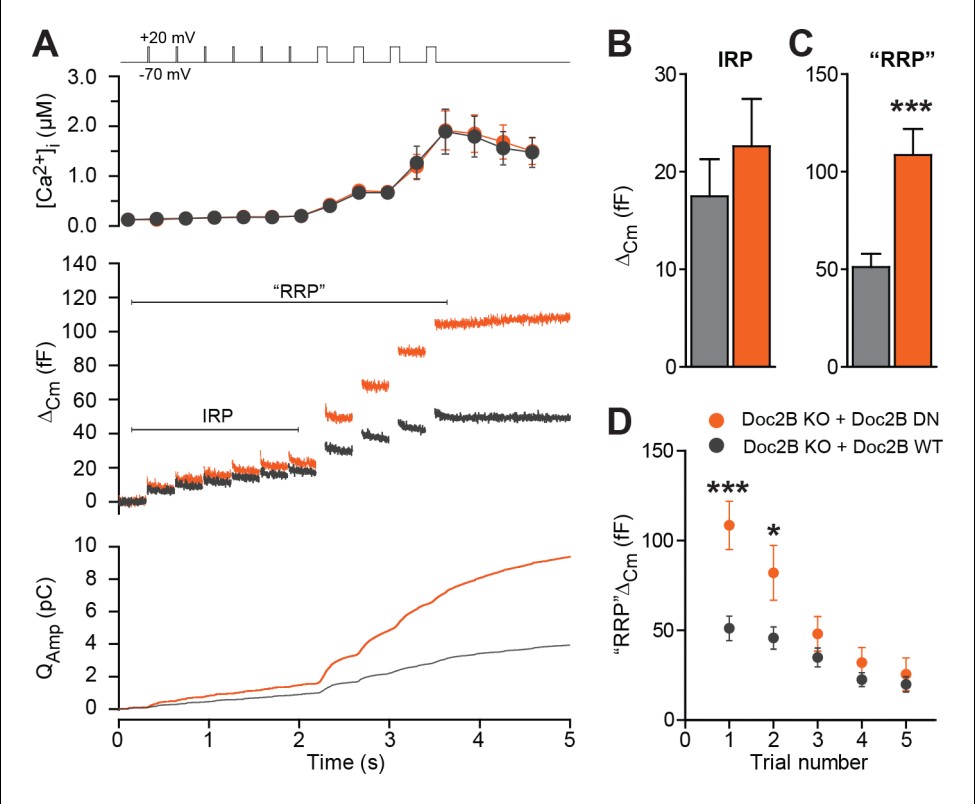

**Figure 6.** The Doc2B DN-mutant promotes the filling of the Readily Releasable vesicle Pool (RRP), but not the Immediately Releasable Pool (IRP). (**A**) A depolarization protocol, consisting of 6 brief (10 ms) and four long (100 ms) depolarizations from −70 mV to +20 mV, was repeated a total of 5 times (trials 1–5), separated by 60 s recovery periods. Only the first trial is depicted. A pipette solution with no added calcium and containing 0.5 mM EGTA was used (n = 21 cells for Doc2B WT; n = 22 cells for Doc2B DN). Top panel shows average (±SEM) $[Ca^{2+}]_i$. Middle panel shows the changes in cell membrane capacitance and bottom panel shows the integrated amperometric current (mean traces for all cells). (**B**) The size of the Immediately Releasable Pool (IRP) was not significantly affected by overexpression of Doc2B DN while (**C**) the size of the RRP was significantly enhanced by the Doc2B DN mutant, compared to Doc2B WT. (**D**), Plot of the mean (±SEM) RRP size over the five trials - the responses decrease because of secretory run-down. *p<0.05, ***p<0.001; unpaired, two-tailed Student's t-test.
DOI: https://doi.org/10.7554/eLife.27000.009

levels similar to WT protein and did not change the staining for the vesicular marker syb-2 (*Figure 2A–D*). Using an EGFP-fusion construct we verified that Doc2B ΔMID trafficked reversibly to the membrane indistinguishable from the WT protein when expressing chromaffin cells were exposed to High-K⁺ (*Figure 3A–C*). This further shows that the C2-domains are correctly folded in this mutant. Strikingly, expression of the ΔMID mutant in Doc2B KO cells led to partial recovery of the sustained component (*Figure 9Ciii*), which reached 42.7 ± 6.7 fF/s whereas expressing Doc2B WT depressed the sustained component to 13.3 ± 2.2 fF/s. Consistent with the results obtained with the mutated MID, the ΔMID also did not support the fast burst amplitude seen with Doc2B WT (*Figure 9Ci*). Although a sustained release rate of 42 fF/s is quite typical for WT chromaffin cells, Doc2B KO cells display on average higher rates (60–70 fF/s; *Figure 9Ciii*), which indicates that other regions outside the MID domain also contribute to the inhibitory effect. Together with the findings above that the negative effects of Doc2B on secretion entirely dominate in the Munc13-2 KO chromaffin cells (*Figure 1D*), this shows that the negative function of Doc2B on release requires the MID-domain, but not Munc13-2 (see also Discussion).

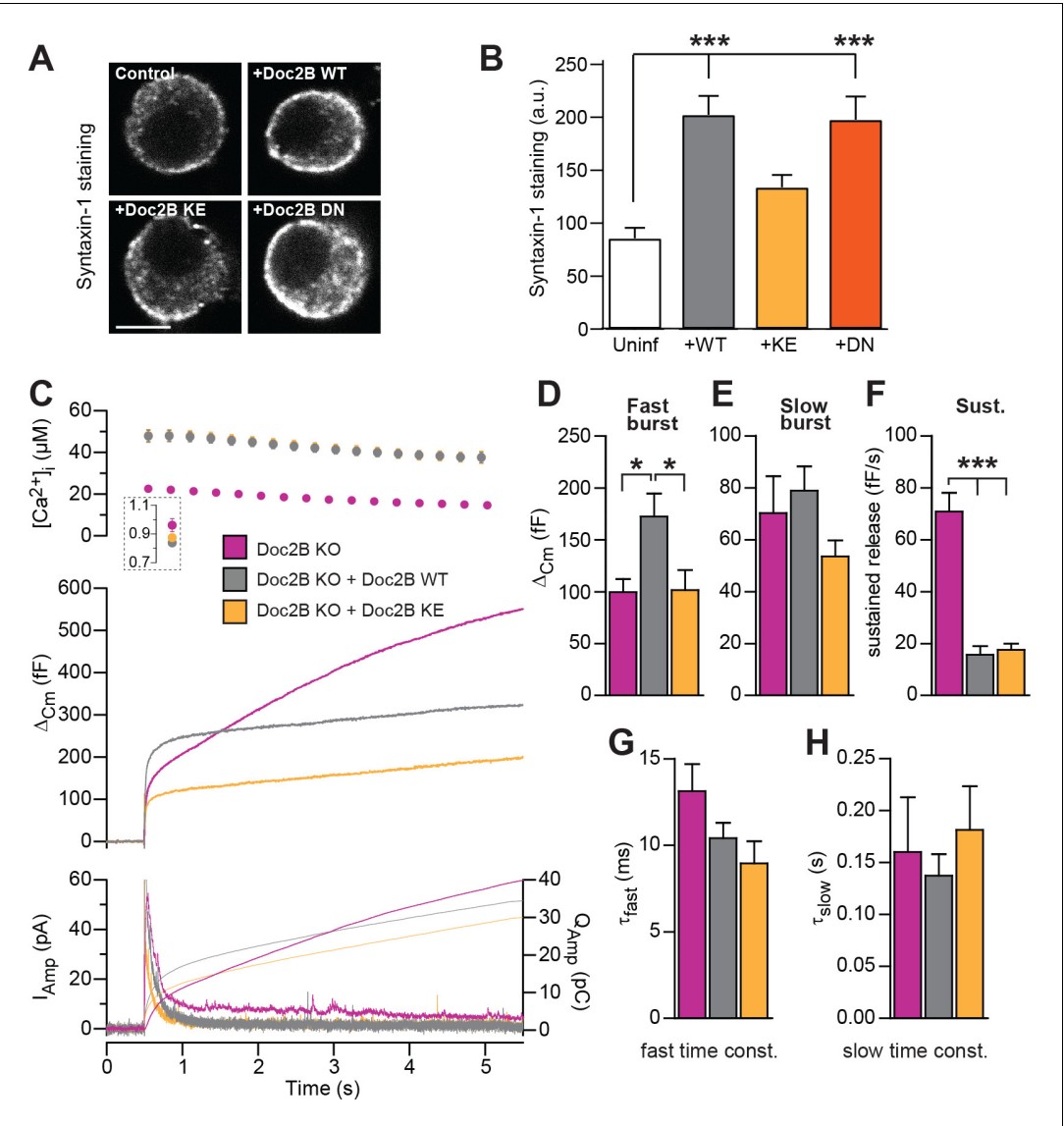

**Figure 7.** SNARE-binding of Doc2B is required for boosting the RRP, but not for inhibiting the sustained phase. (A) Fixed and Immunostained wildtype chromaffin cells stained against syntaxin-1. (B) Quantification of the intensity of syntaxin-1 staining shows that expression of Doc2B WT (n = 12 cells) lead to an increase in intensity compared to non-expressing control cells ('Uninf', n = 11). The KE mutation (K237,319E; n = 15) did not significantly affect syntaxin-1 staining, whereas the DN mutant increased staining (n = 12). ***$p < 0.001$, Dunn's multiple comparison test. (C) $Ca^{2+}$ uncaging experiment in Doc2B KO cells overexpressing Doc2B WT (n = 16) or Doc2B KE (K237,319E; n = 19). Recordings from another experiment in uninfected Doc2B KO cells (pink; same data as in *Figure 9*) are shown in parallel (n = 25); (D-H) Kinetic analysis of the capacitance traces. The size of the fast burst upon Doc2B KE overexpression was similar to Doc2B KO cells and smaller that in cells overexpressing Doc2B WT (D) while the slow burst showed a tendency for reduced size (E). The sustained phase was similarly affected by expression of either Doc2B KE or Doc2B WT (F). No differences were identified in the release time constants (G,H). *$p < 0.05$, ***$p < 0.001$; One-way ANOVA with Tukey's multiple comparison test.
DOI: https://doi.org/10.7554/eLife.27000.010

## Discussion

In chromaffin cells, Doc2B has dual effects: it stimulates the size of the RRP from rest by boosting priming, but it inhibits priming/fusion during sustained $Ca^{2+}$-elevations, leading to a shallower sustained release component. Thus, Doc2B joins the ranks of a number of exocytotic proteins – including complexin, syts and Munc18s – that have positive and negative effects on different phases of

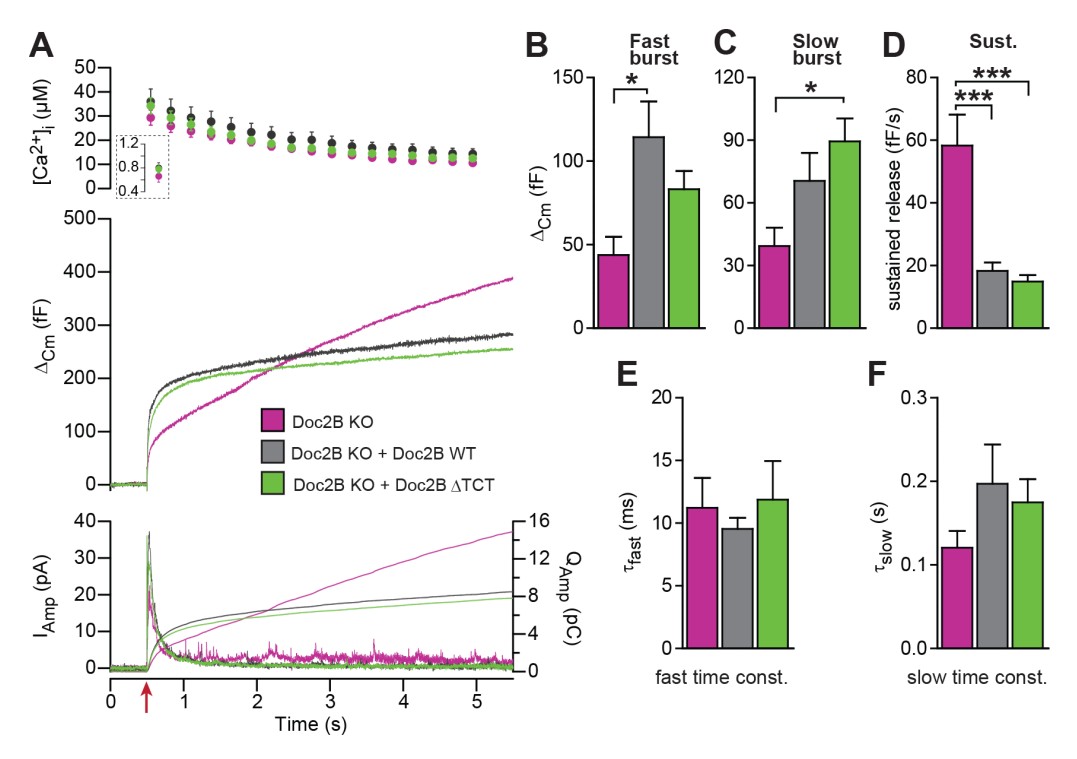

**Figure 8.** A Doc2B mutant lacking the binding sequence for Tctex-1 (Doc2BΔTCT) supports normal secretion. (**A**) Ca²⁺ uncaging experiment in Doc2B KO cells (n = 13) and KO cells overexpressing Doc2B WT (n = 15) or Doc2BΔTCT (n = 14). (**B–F**) Kinetic analysis revealed that both vesicle pool sizes and kinetics of release were not significantly different between Doc2B WT and Doc2BΔTCT. *p<0.05; ***p<0.001, Dunn's multiple comparison test.
DOI: https://doi.org/10.7554/eLife.27000.011

release. Here, we have dissected the involvement of the different Doc2B effectors in this phenotype. Cross-rescue showed that syt-1 and ubMunc13-2 are required for the overall stimulating effect on priming, whereas syt-7 is dispensable. Even in the absence of syt-1 or syt-7, an inhibitory function of Doc2B was seen, and inhibition dominated in the absence of Munc13-2. Mutagenesis showed that interactions with SNAREs, Munc13, and Ca²⁺ are involved in the positive effects of Doc2B on priming, whereas they were not required for the inhibitory function. Thus, the stimulatory and inhibitory effects of Doc2B can be separated by mutation. This rules out that the stimulatory effect on RRP size could be a secondary effect of the inhibition of sustained release, which in principle could lead to a larger RRP by preventing loss of vesicles.

Strikingly, both the DN and 6A mutations rendered the exocytotic burst size independent of basal [Ca²⁺]ᵢ. With the DN mutation, the primed vesicle pools attained their maximal size irrespective of basal [Ca²⁺]ᵢ, occluding Ca²⁺-dependent priming (*Bittner and Holz, 1992*; *Voets, 2000*), whereas – with the 6A mutation – the RRP size was intermediate, but calcium-independent. The increase in secretion induced by the DN-mutant was previously reported by *Friedrich et al. (2008)*, who used expression in WT mouse chromaffin cells; here we show by direct comparison to the wildtype protein in a rescue setting that the DN-mutation renders overall priming calcium-independent. Where the RRP was increased, the IRP was not significantly affected by the DN-mutation (*Figure 6*). The reason for this is unknown, but the IRP forms a subpool of the RRP (*Voets et al., 1999*), which contains vesicles close to Ca²⁺-channels. Other factors (for instance the number or availability of Ca²⁺-channels) might therefore limit the size of the IRP.

The DN-mutation is situated in the C2A-domain (*Groffen et al., 2006*), whereas the C2B-domain also contributes essentially to Ca²⁺-induced membrane trafficking (*Giladi et al., 2013*; *Xue et al., 2015*). The finding that the DN-mutant (and other neutralization of the Ca²⁺-binding ligands in the C2A-domain [*Gaffaney et al., 2014*]) lead to permanent membrane anchoring can be rationalized from the reduced electrostatic repulsion between the DN-mutant and negatively charged membrane

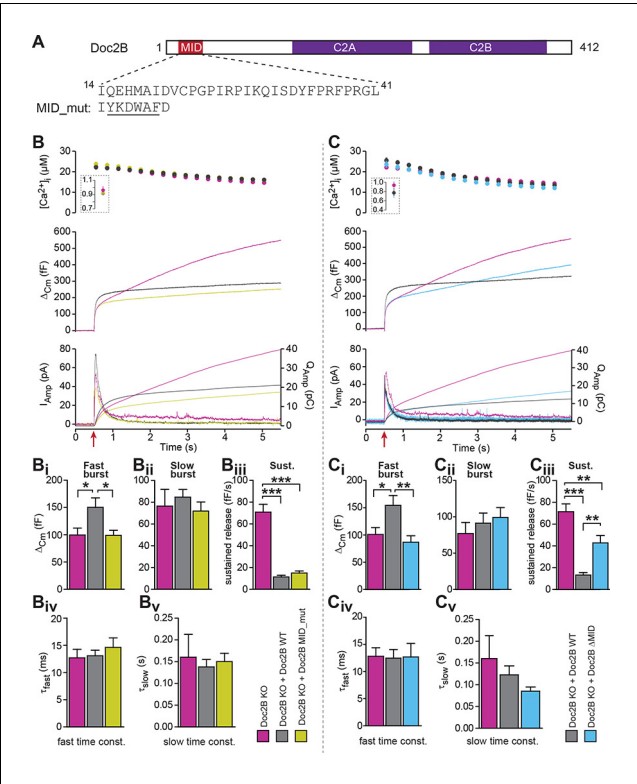

**Figure 9.** The MID-domain is required for both positive and negative functions of Doc2B in vesicle priming. (**A**) Domain structure of mouse Doc2B. C2A and C2B are the two C2-domains. The MID-domain encompasses amino acids 14–38. We mutated the N-terminal part of the MID-domain (MID_mut, (**Mochida et al., 1998**)), or deleted it entirely (deletion of amino acids 14–41: ΔMID). (**B**) $Ca^{2+}$ uncaging experiment in Doc2B KO cells (n = 25) and KO cells overexpressing Doc2B WT (n = 26) or Doc2B MID_mut (n = 26). (**Bi-Bv**) Kinetic analysis showed that mutating a stretch of the Doc2B MID domain abolishes the facilitating effect that Doc2B WT has in potentiating the RRP size (i.e. the size of the fast burst) (**Bi**) while causing a similar reduction in the rate of sustained release as the wildtype protein (**Biii**). (**C**), Full deletion of the Munc13-interacting domain of Doc2B (amino acids 14–41; Doc2B ΔMID) interferes with the function of Doc2B in both the fast burst and the sustained phase of release. $Ca^{2+}$ uncaging experiment in Doc2B KO cells overexpressing Doc2B WT (n = 18) or Doc2B ΔMID (n = 20). (**Ci-Cv**) Kinetic analysis showing that removal of the full MID region of Doc2b interferes both with the promoting action on the filling of the fast pool of vesicles (**Ci**) and alleviates the suppression of the sustained phase of release caused by Doc2B WT overexpression (**Ciii**). The data from recordings of uninfected Doc2B KO cells used in panels C is the same as the one in panels B. *p<0.05, **p<0.01, ***p<0.001; Dunn's multiple comparison test.
DOI: https://doi.org/10.7554/eLife.27000.012

phospholipids (phosphatidylserine and PI(4,5)P$_2$), in the presence of an affinity for the membrane via the C2B-domain. Indeed, recent data show that membrane binding of both WT Doc2B and DN-mutant requires PI(4,5)P$_2$ at the plasma membrane (**Michaeli et al., 2017**). Thus both $Ca^{2+}$-bound Doc2B WT and the DN-mutant localize to similar PI(4,5)P$_2$ containing domains on the plasma membrane.

Whereas mutation of the coordinating aspartates in Doc2B (either DN or 6A mutation) renders vesicle priming $Ca^{2+}$-independent, priming is still $Ca^{2+}$-dependent in the absence of Doc2B (comp. **Figure 4A and G**), as noted before (**Pinheiro et al., 2013**). Therefore, the mutated Doc2B occludes an endogenous $Ca^{2+}$-dependent priming factor. To gain further insight into the function of Doc2B in different priming steps, we considered the 3-pool model for exocytosis, where the RRP is refilled reversibly from the SRP, and the SRP in turn is refilled reversibly from a Depot pool (which may or may not be docked, **Figure 10**). In this model, the overall $Ca^{2+}$-dependence of the burst size is due to a $Ca^{2+}$-dependence of the transition between Depot and SRP (called *priming 1*, **Figure 10**) (**Voets, 2000**; **Walter et al., 2013**), which indirectly affects also the RRP. In our previous work (**Pinheiro et al., 2013**) we concluded that Doc2B acts primarily on the interconversion between SRP

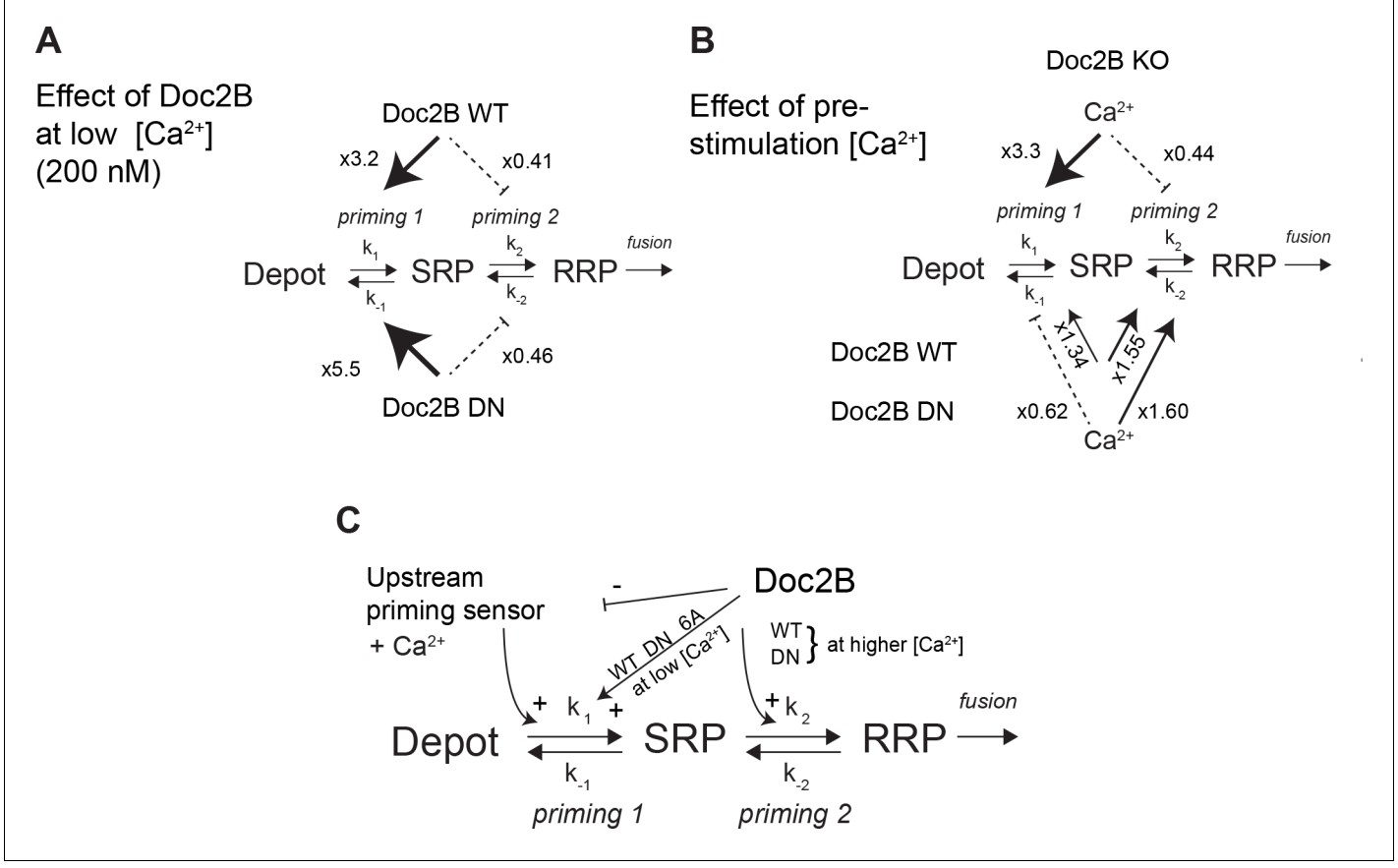

**Figure 10.** Dissecting the effect of Doc2B on two sequential priming steps. (**A**) Chromaffin cells feature two sequential priming steps: *priming 1*, which recruits vesicles to the SRP, and *priming 2*, which recruits them to the RRP. At equilibrium, the overall propensity for forward priming is $k_1[Depot]/k_{-1}$ and $k_2/k_{-2}$, respectively (see Materials and methods). Shown is the multiplicative effect of expressing Doc2B WT and DN mutant on *priming 1* and *priming 2* propensities at low [Ca²⁺]. Both proteins stimulate *priming 1*, but inhibit *priming 2*. It has been proposed that SRP-vesicles fuse directly (**Voets, 2000**), or, as indicated here, that they mature to RRP-vesicles before they fuse (**Walter et al., 2013**); the experiments we present here do not distinguish between two possibilities, but see (**Walter et al., 2013**). (**B**) The effect of increasing [Ca²⁺] from 200 to ~ 800 nM, in the Doc2B KO, or after expression of Doc2B WT or DN. Ca²⁺ stimulates *priming 1* strongly in the Doc2B KO, but inhibits *priming 2*, whereas with Doc2B, *priming 2* is stimulated by Ca²⁺. (**C**) Overall model for the effect of Doc2B on *priming 1* and *priming 2*. Doc2B (and the DN and 6A mutants) stimulates *priming 1*, but also occludes its Ca²⁺-dependence, presumably by inhibiting or competing with another Ca²⁺-dependent priming protein (upstream priming sensor) that normally acts at *priming 1*. Ca²⁺ increases the effect of Doc2B on priming 2; this effect is seen also in the DN-mutant, but not in the 6A-mutant; therefore, it depends on Ca²⁺ binding to the C2B-domain.

DOI: https://doi.org/10.7554/eLife.27000.013

The following source data is available for figure 10:

**Source data 1.** Values for pool sizes and *priming 1* and *priming 2* propensity from experiments in Doc2B KO, and after overexpression of Doc2B WT and DN-mutant (see also *Figure 4*).
DOI: https://doi.org/10.7554/eLife.27000.014

and RRP vesicles (*priming 2* in *Figure 10*). Therefore, in the Doc2B KO, the burst size (SRP + RRP) is still Ca²⁺-dependent (*Figures 4* and *5* and [*Pinheiro et al., 2013*]). By assuming that the steady state has been obtained before we uncage calcium, we are able (from the data in *Figure 4*) to calculate the ratios $k_1[Depot]/k_{-1}$ and $k_2/k_{-2}$, which are the overall propensities of *priming 1* and *priming 2* to drive priming forward, and compare them between genotypes (*Figure 10*, *Figure 10—source data 1*). We considered the effect of having the Doc2B protein present at ~200 nM Ca²⁺ (*Figure 10A*), and the effect of raising pre-stimulation [Ca²⁺]ᵢ from ~200 nM to ~800 nM (*Figure 10B*). The conclusion is that Doc2B acts on both priming steps, but in different ways: it acts on *priming 2* at higher [Ca²⁺] (*Figure 10B*) to drive vesicles to the RRP (*Pinheiro et al., 2013*), but at low [Ca²⁺] it acts to drive *priming 1* forward (*Figure 10A*). This can be appreciated directly from the data by noting the

large increase in SRP size in *Figures 4C* and *5C* when expressing Doc2B or its mutants and probing from low basal [Ca$^{2+}$]. The effect on *priming 1* is even larger for the DN mutant than for the WT protein. The result of the upstream action is that *priming 1* loses some or (in the DN mutant) all of its Ca$^{2+}$-sensitivity, and indeed, the Ca$^{2+}$-sensitivity of *priming 1* is even higher in the total absence of Doc2B than in the presence of WT Doc2B (*Figure 10B*). Most likely there is competition between Doc2B and the upstream Ca$^{2+}$-dependent priming sensor for *priming 1* (*Figure 10C*), and this is affected by mutation and probably by the relative abundance of the two sensors near the membrane. The physiological relevant function of the Doc2B protein can be appreciated from the fact that in the Doc2B KO, Ca$^{2+}$ suppresses the transition from the SRP into the RRP, whereas in the presence of Doc2B, this transition is stimulated by Ca$^{2+}$ (*Figure 10B*). Thus, the upstream Ca$^{2+}$-sensor, when left to its own devices, would push vesicles effectively into the SRP, but the transition into the RRP would be inhibited. Doc2B, on the other hand, can prime vesicles further into the RRP by Ca$^{2+}$-binding. Strikingly, another conclusion is that only the WT protein and the DN-mutant can push the vesicles into the RRP at higher [Ca$^{2+}$] – the 6A-mutant is not able to do that (*Figure 5H*) - which identifies Ca$^{2+}$-binding to the C2B-domain as the decisive event for priming 2 (*Figure 10C*). The identity of the upstream Ca$^{2+}$-dependent priming sensor acting on *priming 1* cannot be identified from this work, but candidates are synaptotagmin-7 (*Liu et al., 2014*) and Munc13 isoforms (*Zikich et al., 2008*; *Lipstein et al., 2012*).

At least two different – but nonexclusive – ideas for the action of Doc2 proteins in exocytotic processes have been put forward. The first idea is that Doc2 proteins participate in a direct capacity during the fusion process itself, by triggering Ca$^{2+}$-dependent lipid insertion of its C2 domains, leading to membrane buckling/distortion, which assists in SNARE-dependent membrane fusion – that is, a function akin to that often ascribed to syts. This notion has support in the observation that Doc2 proteins can accelerate membrane fusion in vitro between populations of liposomes carrying nothing but reconstituted SNAREs (*Groffen et al., 2010*; *Yao et al., 2011*; *Yu et al., 2013*; *Gaffaney et al., 2014*). The other idea is that Doc2 serves mainly as a recruitment scaffold, which brings other factors to the membrane – or to the SNAREs, since SNARE-binding is needed (*Sato et al., 2010*) (and our *Figure 7*) – leading to more indirect, that is, more upstream, effects on fusion. This model has indirect support through the ability of a MID-domain peptide to interfere with synaptic transmission (*Mochida et al., 1998*), and by the observation of Doc2B-dependent Munc13-1 recruitment (*Friedrich et al., 2013*). Munc13-1, in turn, might have direct effects on the fusogenicity of vesicles via SNARE-assembly (*Ma et al., 2013*), or by modulating the size of the fusion barrier (*Basu et al., 2007*). In pancreatic beta cells, a function of Doc2B as a scaffolding platform for Munc18 proteins has been identified (*Jewell et al., 2008*; *Ramalingam et al., 2014*).

Our data obtained in adrenal chromaffin cells add several lines of support to a more indirect mode of Doc2B action: first, in cross-rescue experiments, Doc2B cannot stimulate release in the absence of syt-1, the fast calcium sensor for fusion in chromaffin cells. Second, the main promoting effect of WT Doc2B expression is to increase the size of the RRP, which is an effect upstream of fusion. No consistent effect on the rate constant of release, or its Ca$^{2+}$-sensitivity, was observed by the wildtype or mutated Doc2B protein (with the exception of the DN mutation, see below), see also (*Pinheiro et al., 2013*). This is in stark contrast to mutations reducing the Ca$^{2+}$-affinity of syt-1 (*Sørensen et al., 2003a*), compromising syt-1:SNARE-binding (*Mohrmann et al., 2013*), deletion of complexin (*Dhara et al., 2014*), or interference with the C-terminal zippering of the SNARE-complex (*Sørensen et al., 2006*; *Walter et al., 2010*), which all compromise fusion time constants, reflecting the known function of the syt-1:SNARE:complexin supercomplex in fusion triggering. Third, the main effect of mutating the Ca$^{2+}$ coordinating residues of Doc2B was to render overall vesicle priming Ca$^{2+}$-independent (*Figures 4–5*). Fourth, we find that mutating the MID domain or removing ubMunc13-2 – the major Munc13 isoform in chromaffin cells (*Man et al., 2015*) – abolished the stimulating effect of Doc2B on priming (*Figure 1D*). Overall, therefore, we favor the idea that Doc2B acts through the recruitment of other factors, which cause vesicle priming.

Even as the indirect effects of Doc2B dominate in chromaffin cells, Doc2B appears to have a more direct effect on fusion in other systems, or after mutation. In neurons the DN mutation increases the mEPSC frequency (*Groffen et al., 2010*; *Gaffaney et al., 2014*), but it also increased synchronous release and RRP size (*Gaffaney et al., 2014*; *Xue et al., 2015*), and when expressed in syt-1 KO neurons it supported a larger increase in asynchronous release than the WT protein (*Xue et al., 2015*). When expressed in adrenal chromaffin cells the DN-mutant was the only tested

mutant which decreased the time constant of fast burst fusion (*Figure 4K*), suggesting a function in accelerating fusion itself. These observations would be consistent with the idea that the DN mutation – in addition to locating Doc2B at the membrane – potentiates a direct function of Doc2B in fusing membranes. This function might dominate in neurons because the small synaptic vesicles are intrinsically more fusogenic due to their high curvature (or for other reasons), but only become noticeable in chromaffin cells in the context of the DN-mutant.

The negative effect that Doc2B exerts on sustained release in chromaffin cells (*Figure 1*) is not an artifact of a high expression level, because sustained release is augmented in the Doc2B KO compared to WT cells from littermates (*Pinheiro et al., 2013*), indicating that the endogenous expression level suffices for the effect. The effect can also be seen when stimulating by depolarization from a low resting $[Ca^{2+}]$ (~100 nM) through augmentation of 'delayed release' between stimulations in Doc2B KO cells (*Pinheiro et al., 2013*). Overexpression of Doc2B or its mutants did not change staining for syb2, a vesicular marker (*Figure 2*). This is an important control, because overexpression of Doc2B increases spontaneous release in chemical synapses (*Groffen et al., 2010*). Increased spontaneous release might in principle lead to vesicle depletion and a secondary reduction in sustained release, although this is unlikely as the effect on spontaneous release is modest (a factor of ~2) and did not change RRP size in synapses (*Groffen et al., 2010*). Using electron microscopy, we previously (*Pinheiro et al., 2013*) reported a significant increase in the number of granules upon Doc2B expression. This increase was not reflected in a changed syb2 expression in this manuscript. Electron microscopy and immunostaning are very different methods for assessing vesicle abundance, and further experiments will be required to understand the reason for this apparent discrepancy. The phenotype of increased RRP and inhibited sustained was previously found after overexpression of open syntaxin-1 (*Liu et al., 2010*), which together with our observation that Doc2B recruits syntaxin-1 from plasma membrane clusters (*Toft-Bertelsen et al., 2016*) suggest an explanation for the phenotype. However, the KE-mutation, which did not recruit syntaxin-1, still limited sustained release. The inhibitory function was partly relieved after deletion of the entire MID-domain. Inhibition dominated when Doc2B was expressed in Munc13-2 KO cells, but in these cells the fast and slow bursts of release were also reduced, along with the sustained component. This indicates that vesicle priming is fundamentally different in Munc13-2 KO cells. Priming in chromaffin and PC12 cells depends on Munc13 and CAPS proteins (*Liu et al., 2010*; *Kabachinski et al., 2014*). Munc13-1 requires CAPS-1 to be able to increase secretion in chromaffin cells (*Liu et al., 2010*). The major Munc13 variant in chromaffin cells is ubMunc13-2 (*Man et al., 2015*), but the specific interactions between ubMunc13-2 and CAPS-1/2 – and between ubMunc13-2 and Doc2B – remain essentially unexplored. Interestingly, expressing Munc13-1 in chromaffin cells yields a higher value of the ratio between burst/sustained, ~0.7 at 4 s post-flash, whereas ubMunc13-2 supported a relatively larger sustained component, yielding a ratio between burst/sustained of ~0.4 (*Man et al., 2015*). CAPS-proteins are required for the sustained release component in chromaffin cells (*Liu et al., 2008*; *Liu et al., 2010*) and traffic to the membrane on secretory vesicles (*Kabachinski et al., 2016*); therefore we can hypothesize that ubMunc13-2 displays more efficient coupling to CAPS activity than Munc13-1. It is an attractive possibility that Doc2B might target one Munc13 paralog (Munc13-1) selectively to the release machinery, thus accounting for a larger primed vesicle pool, while sustained release is limited by an additional function of Doc2B to block interactions with CAPS-proteins or the upstream $Ca^{2+}$-sensor for priming (*priming 1*, *Figure 10*).

In conclusion, we show here that localization of Doc2B to the plasma membrane occludes the upstream $Ca^{2+}$-dependent priming step (*Bittner and Holz, 1992*; *von Rüden and Neher, 1993*; *Voets, 2000*), and that Doc2B acts as a calcium sensor for vesicle priming into the RRP (*Figure 10C*). We further demonstrate that this function includes interactions with SNAREs and Munc13-2, and it further requires syt-1 to be present. Another, upstream, $Ca^{2+}$-dependent priming factor is present in chromaffin cells, as vesicle priming remains $Ca^{2+}$-dependent even in the Doc2B KO. Another function of Doc2B in adrenal chromaffin cells is to limit sustained release. This function also depends on the MID-domain, and we tentatively suggest that Doc2B might target some priming factors (Munc13, CAPS) selectively to the exocytotic cascade to shape overall release kinetics. Thus, the end result of Doc2A/B function will depend on the priming factors expressed in a particular cell and therefore will vary between systems. This might account for the different functions of Doc2B in beta cells, (increased biphasic insulin release [*Ramalingam et al., 2012*]), adrenal chromaffin cells (increase in RRP, but decrease in sustained release [*Pinheiro et al., 2013*]), and neurons

(increase in spontaneous and/or asynchronous release [*Groffen et al., 2010*; *Pang et al., 2011*; *Yao et al., 2011*]).

# Materials and methods

## Key resources table

| Reagent type (species) or resource | Designation | Source or reference | Identifiers | Additional information |
|---|---|---|---|---|
| strain, strain background (M. musculus) | C57BL/6 | | | |
| strain, strain background (M. musculus) | CD1 | | | |
| genetic reagent (M. musculus) | Doc2a null allele | Sakaguchi G, Manabe T, Kobayashi K, Orita S, Sasaki T, Naito A, Maeda M, Igarashi H, Katsuura G, Nishioka H, Mizoguchi A, Itohara S, Takahashi T, Takai Y. Doc2alpha is an activity-dependent modulator of excitatory synaptic transmission. Eur J Neurosci. 1999 Dec;11(12):4262–8. | PMID: 10594652 | |
| genetic reagent (M. musculus) | Doc2b null allele | Groffen AJ, Martens S, Díez Arazola R, Cornelisse LN, Lozovaya N, de Jong AP, Goriounova NA, Habets RL, Takai Y, Borst JG, Brose N, McMahon HT, Verhage M. Doc2b is a high-affinity Ca2+ sensor for spontaneous neurotransmitter release. Science. 2010 Mar 26;327 (5973):1614–8. | PMID: 20150444 | |
| genetic reagent (M. musculus) | Synaptotagmin-7 (syt7) null allele | Maximov A, Lao Y, Li H, Chen X, Rizo J, Sørensen JB, Südhof TC. Genetic analysis of synaptotagmin-7 function in synaptic vesicle exocytosis. Proc Natl Acad Sci U S A. 2008 Mar 11;105(10):3986–3991. | PMID: 18308933 | |
| genetic reagent (M. musculus) | Synaptotagmin-1 (syt1) null allele | Geppert M, Goda Y, Hammer RE, LI C, Rosahl TW, Stevens CF, Südhof TC. 1994. Synaptotagmin I: a major Ca2 + sensor for transmitter release at a central synapse. Cell 79(4): 717–727. | PMID: 7954835 | |
| genetic reagent (M. musculus) | Unc13b null allele | Varoqueaux F, Sigler A, Rhee J-S, Brose N, Enk C, Reim K, Rosenmund C. Total arrest of spontaneous and evoked synaptic transmission but normal synaptogenesis in the absence of Munc13-medicated vesicle priming. Proc Natl Acad Sci U S A. 2002. Jun 25; 99(13):9037–9042. | PMID: 12070347 | |
| transfected construct (R. norvegicus) | pSFV-Doc2b-WT-IRES-EGFP | Friedrich R, Groffen AJ, Connell E, van Weering JR, Gutman O, Henis YI, Davletov B, Ashery U. DOC2B acts as a calcium switch and enhances vesicle fusion. J Neurosci. 2008 Jul 2;28(27):6794–806. | Genbank Accession: EU635444.1 | local reference: SG#334 |
| transfected construct (R. norvegicus) | pSFV-Doc2b-DN-IRES-EGFP | Friedrich R, Groffen AJ, Connell E, van Weering JR, Gutman O, Henis YI, Davletov B, Ashery U. DOC2B acts as a calcium switch and enhances vesicle fusion. J Neurosci. 2008 Jul 2;28(27):6794–806. | Genbank Accession: EU635445.1 | derived from EU635444.1 with indicated mutations |
| transfected construct (R. norvegicus) | pSFV-Doc2b-6A-IRES-EGFP | this paper | | derived from EU635444.1 with indicated mutations |

*Continued on next page*

*Continued*

| Reagent type (species) or resource | Designation | Source or reference | Identifiers | Additional information |
|---|---|---|---|---|
| transfected construct (*R. norvegicus*) | pSFV-Doc2b-KE-IRES-EGFP | this paper | | derived from EU635444.1 with indicated mutations |
| transfected construct (*R. norvegicus*) | pSFV-Doc2b-TCT-IRES-EGFP | this paper | | derived from EU635444.1 with indicated mutations |
| transfected construct (*R. norvegicus*) | pSFV-Doc2b-MID (14-41)-IRES-EGFP | this paper | | derived from EU635444.1 with indicated mutations |
| transfected construct (*R. norvegicus*) | pSFV-Doc2b-MID (Scrambled)-IRES-EGFP | this paper | | derived from EU635444.1 with indicated mutations |
| transfected construct (*R. norvegicus*) | pLenti-Doc2bEGFP-WT | this paper | | derived from EU635444.1; local reference: J274#2 |
| transfected construct (*R. norvegicus*) | pLenti-Doc2bEGFP-DN | this paper | | local reference: J274#7 |
| transfected construct (*R. norvegicus*) | pLenti-Doc2bEGFP-6A | this paper | | local reference: K102#33 |
| transfected construct (*R. norvegicus*) | pSFV-EGFP-Doc2B-DeltaMID | this paper | | local reference: #480 |
| transfected construct () | | | | |
| transfected construct () | | | | |
| transfected construct () | | | | |
| biological sample () | | | | |
| antibody | anti-syntaxin | Synaptic System | SySy: 110011 | dil. 1:1000; Overnight/room temperature |
| antibody | anti-GFP | Abcam | Ab13970 | dil. 1:1000; Overnight/room temperature |
| antibody | secondary Goat anti-mouse Alexa 488 | Invitrogen | Invitrogen: A11029 | dil. 1:1000; Overnight/4 deg. |
| antibody | secondary Goat anti-chicken Alexa 488 | Abcam | Ab150169 | dil. 1:1000; Overnight/4 deg. |
| antibody | anti-synaptotagmin-1 | Synaptic System | SySy: 105011 | dil. 1:1000; Overnight/4 deg. |
| antibody | anti-synaptotagmin-7 | Synaptic System | SySy: 105173 | dil. 1:500; Overnight/4 deg. |
| antibody | anti-VCP | Abcam | Ab11433 | dil. 1:2000; 1h/room temperature |
| antibody | mouse mAb against Syb2, clone 69.1 | Synaptic Systems GmbH | cat#104211 | 1:2500 diluted in PBS+ 2% BSA, incubated 2 hr at RT |

*Continued*

| Reagent type (species) or resource | Designation | Source or reference | Identifiers | Additional information |
|---|---|---|---|---|
| antibody | rabbit pAb against Doc2b | Groffen AJ, Brian EC, Dudok JJ, Kampmeijer J, Toonen RF, Verhage M. Ca(2+)-induced recruitment of the secretory vesicle protein DOC2B to the target membrane. J Biol Chem. 2004 May 28;279(22):23740–7. | PMID: 15033971 | 1:133 diluted in PBS+ 2% BSA, incubated 2 hr at RT (note: Doc2b and Syb2 stainings were performed separately on different coverslips to rule out potential spectral overlap)/For western blotting: dil. 1:500; Overnight at 4 deg |
| antibody | Goat anti-rabbit HRP | Agilent | Dako-P0448 | dil. 1:10000; 1h30/room temperature |
| antibody | Goat anti-mouse HRP | Agilent | Dako-P0447 | dil. 1:10000; 1h30/room temperature |
| antibody | Goat-anti-Rabbit Alexa546 | ThermoFisher Scientific | cat#A-11010 | 1:1000 diluted in PBS+ 2% BSA |
| antibody | Goat-anti-Mouse Alexa647 | ThermoFisher Scientific | cat#10739374 | 1:1000 diluted in PBS+ 2% BSA |
| antibody | | | | |
| recombinant DNA reagent | | | | |
| sequence-based reagent | | | | |
| peptide, recombinant protein | | | | |
| commercial assay or kit | BCA Protein assay kit | Pierce | Pierce: 23227 | |
| commercial assay or kit | | | | |
| commercial assay or kit | | | | |
| chemical compound, drug | NaCl | Sigma-aldrich | Sigma-aldrich: S9888 | |
| chemical compound, drug | KCl | Sigma-aldrich | Sigma-aldrich: P5405 | |
| chemical compound, drug | NaH2PO4 | Sigma-aldrich | Sigma-aldrich: S8282 | |
| chemical compound, drug | Glucose | Sigma-aldrich | Sigma-aldrich: G8270 | |
| chemical compound, drug | DMEM | Gibco | Gibco: 31966047 | |
| chemical compound, drug | L-cysteine | Sigma-aldrich | Sigma-aldrich: C7352 | |
| chemical compound, drug | CaCl2 | Sigma-aldrich | Sigma-aldrich: 499609 | |
| chemical compound, drug | EDTA | Sigma-aldrich | Sigma-aldrich: E5134 | |
| chemical compound, drug | papain | Worthington Biochemical | Worthington Biochemical: LS003126 | |
| chemical compound, drug | albumin | Sigma-aldrich | Sigma-aldrich: A3095 | |
| chemical compound, drug | trypsin-inhibitor | Sigma-aldrich | Sigma-aldrich: T9253 | |
| chemical compound, drug | penicillin/ streptomycin | Invitrogen | Invitrogen: 15140122 | |
| chemical compound, drug | insulin-transferrin-selenium-X | Invitrogen | Invitrogen: 51500056 | |
| chemical compound, drug | fetal calf serum | Invitrogen | Invitrogen: 10500064 | |
| chemical compound, drug | MgCl2 | Sigma-aldrich | Sigma-aldrich: 449172 | |
| chemical compound, drug | HEPES | Sigma-aldrich | Sigma-aldrich: H3375 | |
| chemical compound, drug | Nitrophenyl-EGTA (NPE) | Synthesized at the Max-Planck-Institut for biophycial chemistry, Göttingen. | | |

*Continued on next page*

*Continued*

| Reagent type (species) or resource | Designation | Source or reference | Identifiers | Additional information |
|---|---|---|---|---|
| chemical compound, drug | Fura-4F | Invitrogen | Invitrogen: F14174 | |
| chemical compound, drug | Furaptra | Invitrogen | Invitrogen: M1290 | |
| chemical compound, drug | Mg-ATP | Sigma-aldrich | Sigma-aldrich: A9187 | |
| chemical compound, drug | GTP | Sigma-aldrich | Sigma-aldrich: G8877 | |
| chemical compound, drug | Vitamin C | Sigma-aldrich | Sigma-aldrich: A5960 | |
| chemical compound, drug | EGTA | Sigma-aldrich | Sigma-aldrich: E4378 | |
| chemical compound, drug | Paraformaldehyde | Sigma-aldrich | Sigma-aldrich: P6148 | |
| chemical compound, drug | PIPES | Sigma-aldrich | Sigma-aldrich: 80635 | |
| chemical compound, drug | Triton X-100 | Sigma-aldrich | Sigma-aldrich: T8787 | |
| chemical compound, drug | BSA | Sigma-aldrich | Sigma-aldrich: A4503 | |
| chemical compound, drug | Prolong Gold | Invitrogen | Invitrogen: P36934 | |
| chemical compound, drug | Protease cocktail inhibitor | Invitrogen | Invitrogen: 87785 | |
| chemical compound, drug | RIPA buffer | Invitrogen | Invitrogen: R0278 | |
| chemical compound, drug | ECL plus western blotting substrate | Pierce | Pierce: 32132 | |
| chemical compound, drug | | | | |
| chemical compound, drug | | | | |
| chemical compound, drug | | | | |
| chemical compound, drug | | | | |
| software, algorithm | Igor | wavemetrics | | |
| software, algorithm | ImageJ | NIH software | | |

## Chromaffin cell culture

Doc2B knockout (KO) animals were generated by crossing heterozygous mice. Doc2A/B double knockout mice were kept homozygous knockout for both alleles. P0-P1 pups of either sex were sacrificed by decapitation, the adrenal glands dissected out, and chromaffin cells isolated and cultured according to previously published protocols (*Sørensen et al., 2003b*). Briefly, dissected adrenal glands were placed in filtered Locke's solution (154 mM NaCl, 5.6 mM KCl, 0.85 mM $NaH_2PO_4$, 2.15 mM $Na_2HPO_4$, and 10 mM glucose, pH 7.0), and cleaned for connective tissue using tweezers. The glands were digested in 0.3 mL of papain solution (solutions are defined below) at 37°C for 40 min followed by the addition of 0.3 mL of inactivating solution for 5–10 min. This solution was then replaced by 160 µL of enriched DMEM, and the glands triturated through a 200 µL pipette tip. 50 µL of the cell suspension were plated as a drop on glass coverslips in 6-well plates, and the cells were allowed to settle for 20–40 min before adding 2 mL of enriched DMEM. The cells were incubated at 37°C and 8% $CO_2$ and used within 4 days. Papain solution: DMEM (Gibco) supplemented with 0.2 mg/mL L-cysteine, 1 mM $CaCl_2$, 0.5 mM EDTA, and 20–25 U/mL papain (Worthington Biochemical) and equilibrated with 8% $CO_2$. Inactivating solution: DMEM supplemented with 10% heat-inactivated fetal calf serum (Invitrogen), 2.5 mg/mL albumin, and 2.5 mg/mL trypsin inhibitor (Sigma-Aldrich). Enriched DMEM: DMEM supplemented with 4 µL/mL penicillin/streptomycin (Invitrogen) and 10 µL/mL insulin-transferrin-selenium-X (Invitrogen).

## Expression constructs

For short-term re-expression experiments, full-length WT Doc2B, or Doc2B containing specific mutations, were cloned into a Semliki Forest Virus expression vector (pSFV1) containing an Internal Ribosomal Entry Site (IRES) followed by EGFP, to allow for the simultaneous, but independent, expression of both proteins. Fusion-constructs between Doc2B WT, DN and 6A mutants and EGFP (fused to the C-terminus of Doc2B) were cloned in a lentiviral vector. All constructs were verified by sequencing. The generation of lentiviral and SFV particles followed standard protocols.

## Live imaging experiments

Mouse chromaffin cells from CD1 animals (P0) were isolated and plated as described above. After 24 hr, the cells were infected with lentiviral constructs expressing either Doc2b wild-type, the DN mutant or the 6A mutant and all fused C-terminally to the EGFP in order to follow their trafficking in live imaging. For Doc2b ΔMID, 48 hr after primary culture, the cells were infected for 6 hr with a Semliki Forest Virus containing doc2b ΔMID fused to EGFP. After 6 (Semliki Forest virus) or 72 hr (lentiviruses) of infection, confocal imaging of GFP fluorescence (single optical section around the equatorial plane) was performed on a Zeiss LSM780 (488 nm, Argon laser) scanning confocal microscope using a Zeiss Plan-Apochromat 63x/NA 1.4 DIC M27 oil immersion objective (Carl Zeiss,Germany). First acquisition (control) was performed in the presence of extracellular solution (in mM: 145 NaCl, 2.8 KCl, 2 CaCl$_2$, 1 MgCl$_2$, 10 HEPES and 11 glucose, pH 7.2 (osmolarity adjusted to ~305 mOsm)) and imaging was repeated every minute after the application of elevated K$^+$ solution (In mM: 88 NaCl, 59 KCl, 2 CaCl$_2$, 1 MgCl$_2$, 10 HEPES and 11 glucose, pH 7.2 (osmolarity adjusted to ~305 mOsm)) for 5 min ('high K$^+$') and finally every minute after washing with extracellular solution ('wash') for 5 min.

## Exocytosis measurements

Combined capacitance measurements and amperometric recordings were performed as described previously (*Mohrmann et al., 2010*) on a Zeiss Axiovert 10 equipped with an EPC-9 amplifier (HEKA Elektronik) for patch-clamp capacitance measurements and an EPC-7 plus (HEKA Elektronik) for amperometry. The release of catecholamines was triggered by UV flash-photolysis (JML-C2, Rapp Optoelektronik) of a caged-calcium compound, nitrophenyl-EGTA, which was infused into the cell via the patch pipette, or by membrane depolarization.

Intracellular calcium concentrations were determined using a mixture of two fluorescent dyes with different Ca$^{2+}$-affinities [fura-4F and furaptra, Invitrogen; (*Voets, 2000*; *Sørensen et al., 2002*)] which, after calibration, allow calcium concentration measurements over a large dynamic range. For the ratiometric determination of the [Ca$^{2+}$]$_i$ the excitation light (monochromator IV, TILL Photonics, Gräfelfing, Germany) was alternated between 350 nm and 380 nm. The emitted fluorescence was detected with a photodiode and sampled using Pulse software (HEKA Elektronik, Lambrecht/Pfalz, Germany), which was also used for controlling the voltage in the pipette and performing capacitance measurements. The 350/380 fluorescence signal ratio was calibrated by infusion of cells with eight different solutions of known calcium concentrations. The standard intracellular solution contained (in mM): 100 Cs-glutamate, 8 NaCl, 4 CaCl$_2$, 32 Cs-HEPES, 2 Mg-ATP, 0.3 GTP, 5 NPE, 0.4 fura-4F, 0.4 furaptra, and 1 Vitamin C (to prevent UV-induced damage to the fura dyes), pH 7.2 (osmolarity adjusted to ~295 mOsm). For depolarization experiments, Ca$^{2+}$ and NPE were omitted, and EGTA (0.5 mM) was added to the intracellular solution to keep a low basal [Ca$^{2+}$]$_i$. The extracellular solution was composed of (in mM): 145 NaCl, 2.8 KCl, 2 CaCl$_2$, 1 MgCl$_2$, 10 HEPES and 11 glucose, pH 7.2 (osmolarity adjusted to ~305 mOsm).

Amperometric recordings were performed using carbon fiber electrodes with 5 µm in diameter (Thornel P-650/42; Cytec, NJ, USA), insulated using the polyethylene method (*Bruns, 2004*). Amperometry fibers were clamped to 700 mV, and currents were low-pass filtered with a 7-pole Bessel filter at 1 kHz using an EPC-7 plus (HEKA) and sampled at 11.5 kHz.

## Kinetic analysis and priming propensities

Pool sizes and fusion rates were determined by fitting individual capacitance traces with a sum of three exponential functions, each of the form A(1-exp(-t/τ)), representing the fast burst (time constant, τ, 10–30 ms), the slow burst (time constant, τ, 70–300 ms) and the sustained component (time constant, τ, >1 s), respectively, using procedures written in IGOR Pro software. Since the time constant of the sustained component often approached or exceeded the measurement time (5 s), only the mean linear rate of sustained release within 5 s was reported.

To understand the propensities for forward priming in the two sequential priming steps, *priming 1* and *priming 2* (*Figure 10*), we solved the two differential equations

$$\frac{d\mathrm{SRP}}{dt} = k_1\mathrm{Depot} + k_{-2}\mathrm{RRP} - (k_{-1} + k_2)\mathrm{SRP}$$

$$\frac{d\text{RRP}}{dt} = k_2\text{SRP} - k_{-2}\text{RRP}$$

under steady-state assumptions (i.e. the left-hand side were put equal to zero), which yields

$$\frac{k_1\text{Depot}}{k_{-1}} = \text{SRP}_0$$

$$\frac{k_2}{k_{-2}} = \frac{\text{RRP}_0}{\text{SRP}_0}$$

where $\text{RRP}_0$ and $\text{SRP}_0$ are the steady-state values of the two pool sizes. We estimated the $\text{RRP}_0$ and $\text{SRP}_0$ from the response to $Ca^{2+}$-uncaging; the underlying assumptions are that pools are at equilibrium, that the product of $k_1$ and the Depot size is a constant and that there was no ongoing fusion before stimulation.

## Immunocytochemistry

For immunostaining against Doc2B and VAMP2, chromaffin cells were prepared from Doc2A/B double-knockout pups at postnatal day P1. Doc2A/B double knockouts were used for these experiments due to availability, but note that Doc2A is not expressed in adrenal chromaffin cells (*Friedrich et al., 2008*; *Pinheiro et al., 2013*). Eight h post infection with Semliki Infective Particles encoding Doc2b-IRES-EGFP, Doc2B mutants –IRES-EGFP or IRES-EGFP alone as a control, the cells were fixed by adding an equal volume of 4% PFA in PBS to the existing culture medium, yielding a final concentration of 2% PFA. The cells were washed in PBS, permeabilized for 5 min at RT with 0.1% Triton-X100, washed again with PBS, and incubated in PBS containing rabbit polyclonal antiserum 13.2 against Doc2B (*Groffen et al., 2004*) (diluted 1:133) or monoclonal mouse anti-VAMP2 (Synaptic Systems, clone 69.1, diluted 1:2000) for 2 hr at room temperature. The cells were washed 3x for 10 min with PBS and incubated with secondary goat-anti-mouse-A647 and goat-anti-rabbit-A546 (Molecular probes, both diluted 1:1000) for 2.5 hr at RT. The cells were washed 3x for 30 min and mounted in Mowiol. Confocal imaging of A647, A546 and GFP fluorescence was performed on a Zeiss LSM510 microscope using a 63x oil immersion objective.

For immunostaining against syntaxin-1, cells were first prefixed for 5 min in isosmotic fixative containing 0.9% PFA in 0.05 M 1,4-piperazinediethanesulfonic acid (PIPES)/NaOH (pH 7.4, 310 mOsm) 2.8 mM KCl and 1 mM MgCl. This was followed by 8 min of incubation in 3.7% PFA in 0.05 M PIPES/NaOH (pH 7.4). Afterward, the cells were washed three times with PBS (pH 7.4) and permeabilized for 10 min in PBS containing 0.1% Triton X-100 (Sigma-Aldrich), followed by 30 min of blocking in PBS containing 3% BSA. The blocking solution was used for diluting antibodies. The chromaffin cells were incubated overnight with a cocktail of anti-syntaxin 1 and anti-GFP antibodies (1:1000; anti-syntaxin-1, cat. no. 110011; Synaptic Systems and chicken anti-EGFP, cat no. ab13970; Abcam) at room temperature, washed three times with PBS, and incubated with secondary goat anti-mouse Alexa 546 and goat anti-chicken Alexa 488 antibodies (1:1000: Life Technologies) overnight at 4°C. After extensive washings with PBS, cells were rinsed with water and mounted on microscopic slides with Prolong Gold (Invitrogen). For this series of experiments, confocal images were acquired with a HCX PL APO CS lens (NA 1.4; Leica Microsystems A/S) with frame size of 512 × 512 and a zoom factor 10 on a LEICA SP5-X Confocal microscope (Leica Microsystems). Images were corrected for background fluorescence by subtracting an average intensity value of an area not containing any cells. Fluorescence levels of the plasma membrane were quantified by measuring the integrated intensity of a ROI1, defined as the outer edge of the plasma membrane, and subtracting the integrated density of a ROI 2, defined as the inner edge of the membrane. The increase in syntaxin-1 staining by Doc2B expression is more modest using this isosmotic fixation method than with standard PFA-fixation (*Toft-Bertelsen et al., 2016*).

## Electrophoresis and immunoblotting

Adrenal glands were collected from P0-2 Syt-7 oe E18 Syt-1 mice and lysed in RIPA buffer (Invitrogen) supplemented with Protease Inhibitor Cocktail (Invitrogen). The protein concentrations were determined by use of a BCA Protein Assay Kit (Pierce; cat.no.23227). 20 µg (from adrenal glands

samples) of protein were resolved by SDS-PAGE, transferred to nitrocellulose and blotted using anti Doc2B (rabbit pAb 13.2 from Matthijs Verhage laboratory, dil. 1:500), Synaptotagmin-1 (mouse, Synaptic System, Cat. no. SYSY 105011, dil. 1:1000), Synaptotagmin-7 (rabbit, Synaptic System, Cat. no. SYSY 105173, dil. 1:500), or VCP (as loading control, mouse, Abcam, Cat. no. ab11433, dil. 1:2000) antibodies. The blots were developed by chemiluminescence using the ECL plus western blotting substrate system (Pierce). Immunoreactive bands were detected using the FluorChemE image acquisition system (ProteinSimple) and quantified with ImageJ 1.47q (National Institutes of Health).

## Statistics

Data are presented as mean ±SEM, with $n$ indicating the number of cells, and statistical comparisons were made in Graphpad Prism. For every mutant or overexpression condition, we performed control experiments (non-expressing or Doc2B WT expressing) and overexpression/mutant experiments in parallel using cells isolated from the same animals. When testing two conditions, we used a F-test to test for differences in variance. In case the F-test was significant, we performed a Mann-Whitney test, otherwise we used an unpaired two-tailed Student's t-test. In case of three conditions measured in parallel, we tested deviations from homoscedasticity using a Bartlett's test; if the test was significant, we used a Kruskal-Wallis test and Dunn's test for multiple comparison, otherwise we used an ANOVA and Tukey's post-hoc test. For live imaging experiments (*Figure 3C*), we performed a one-sample t-test comparing the change in fluorescence ratio upon stimulation to zero.

## Acknowledgements

We would like to thank Anne-Marie Nordvig Petersen for expert technical assistance. This investigation was supported by the The Lundbeck Foundation (to JBS), the Novo Nordic Foundation (JBS), the Danish Medical Research Council (JBS and SH), and the European Research Council (ERC-ADG-322966-DCVfusion, to MV).

## Additional information

### Competing interests

Iwona Ziomkiewicz: Performed experiments as an employee of University of Copenhagen and is now an employee of AstraZeneca. Has no financial investments in AstraZeneca. The other authors declare that no competing interests exist.

### Funding

| Funder | Grant reference number | Author |
|---|---|---|
| Danish Medical Research Council | | Sébastien Houy<br>Jakob Balslev Sørensen |
| European Research Council | ERC-ADG-322966-DCVfusion | Matthijs Verhage |
| Lundbeckfonden | | Jakob Balslev Sørensen |
| Novo Nordisk Foundation | | Jakob Balslev Sørensen |
| University of Copenhagen | 2016 (KU2016) excellence program | Jakob Balslev Sørensen |

The funders had no role in study design, data collection and interpretation, or the decision to submit the work for publication.

### Author contributions

Sébastien Houy, Formal analysis, Funding acquisition, Investigation, Writing—review and editing; Alexander J Groffen, Iwona Ziomkiewicz, Formal analysis, Investigation, Writing—review and editing; Matthijs Verhage, Conceptualization, Writing—review and editing; Paulo S Pinheiro, Conceptualization, Formal analysis, Funding acquisition, Investigation, Writing—review and editing; Jakob Balslev

Sørensen, Conceptualization, Supervision, Funding acquisition, Writing—original draft, Project administration, Writing—review and editing

## Author ORCIDs
Sébastien Houy http://orcid.org/0000-0003-3639-1931
Jakob Balslev Sørensen https://orcid.org/0000-0001-5465-3769

## Ethics
Animal experimentation: Permission to keep and breed knockout mice for this study was obtained from The Danish Animal Experiments Inspectorate (2006/562−43, 2012−15−2935−00001). The animals were maintained in an AAALAC-accredited stable in accordance with institutional guidelines as overseen by the Institutional Animal Care and Use Committee (IACUC).

## Decision letter and Author response
Decision letter https://doi.org/10.7554/eLife.27000.017
Author response https://doi.org/10.7554/eLife.27000.018

## Additional files
### Supplementary files
• Transparent reporting form
DOI: https://doi.org/10.7554/eLife.27000.015

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
