## [Decision Letter]

Thank you for submitting your article "Doc2B acts as a calcium sensor for vesicle priming requiring synaptotagmin-1, Munc13-2 and SNAREs" for consideration by *eLife*. Your article has been reviewed by two peer reviewers, and the evaluation has been overseen by a Reviewing Editor and Randy Schekman as the Senior Editor. The reviewers have opted to remain anonymous.

The reviewers have discussed the reviews with one another and the Reviewing Editor has drafted this decision to help you prepare a revised submission

Summary:

The manuscript by Houy et al. addresses the role of the Ca^2+^-binding protein Doc2B in secretion from chromaffin cells. By this, the manuscript lines up as a highly significant study in a row of important contributions over the last years studying the function of this protein in secretion from either neurons or neuroendocrine cells. For example, previous work has implicated Doc2B as a Ca^2+^-sensor for spontaneous (Groffen et al., 2010) and asynchronous neurotransmitter release (Yao et al., 2011). In addition, Do2B binds to Munc13-proteins and the SNAREs. Most experiments assess exocytosis by capacitance measurements and amperometry. Exocytosis is stimulated in these experiments by uncaging intracellular Ca^2+^ in the patch clamp whole cell configuration. The experiments are well designed, and the overall quality of the data is excellent.

The results show that Doc2B increases the exocytotic burst and thus the readily releasable pool but it also inhibits the sustained component, which is generally thought to reflect the recruitment followed by the release of unprimed vesicles. They conclude that the action of Doc2B involves Ca^2+^ binding and associated translocation to the plasma membrane and that mutations that do not bind Ca^2+^ and are permanently associated with the plasma membrane are constitutively active, independent of Ca^2+^ binding. In contrast, no changes in Ca^2+^-dependent priming could be detected in Doc2B null mutants (Figure 2 and Figure 4, see also Pinheiro et al., 2013). The authors interpret these observations with the mutant proteins as evidence for Doc2B being a Ca^2+^-sensor for vesicle priming in chromaffin cells. By studying the effect of Doc2B expression in sytI KO and ub-Munc13-2 KO cells and by using specific mutant proteins (Doc2B KE mutant, K237E/K319E) they find that Doc2B action on priming relies on the presence of sytI, Munc13-2 and SNARE binding activity.

Essential revisions:

Reviewer 1:

While the manuscript contains very interesting data leading to novel conclusions some data are less clear and/or not clearly presented, which need to be addressed. Also, the Discussion section is rather confusing. The discussion should specifically address points 2, 4, and 9, listed below.

1) In Figure 1 an important control needs to be included, the overexpression of Doc2B in WT chromaffin cells that are not devoid of endogenous Doc2B background. This is important for assessment of the data in the Syt-1, Syt-7 and Munc13-2 KO cells, which presumably do express WT level Doc2B.

2) Figure 1 legend states overexpression. How do Doc2B expression levels in Doc2B KO cells compare quantitatively with Doc2B levels in WT chromaffin cells? The authors do not convincingly address the possibility that Doc2B abundance may be responsible for inhibition of the sustained phase and the reported effects seen in Syt-1 KO and Munc13-2 KO cells (Figure 1). Syt-1 KO and Munc13-2 KO cells contain endogenous proteins and the overexpression of Doc2B WT may have introduced significantly more protein above background (this is unknown because this data is not included but pSFV is known to induce very high overexpression levels). In case of higher Doc2B protein expression in Syt1 KO+Doc2B WT versus Syt1 KO, there is a possibility that this difference may explain the observed decrease. The same could be said for the Munc13-2 KO experiment (Figure 1).

3) Subsection “Doc2B requires synaptotagmin-1 and Munc13-2 to stimulate priming”, Figure 1Bii: The authors state that in absence of Syt-1, Doc2B has no effect on slow burst. However, the slow burst in syt1 KO is almost 50% larger than that in syt1 KO + Doc2B WT although the error bars are too large to achieve apparently p<0.05 significance. Is this due to the smaller sample size? In Figure 1Ai, similar amount of difference between Doc2B KO and Doc2B KO+Doc2B WT is statistically significant, and the sample size is also larger than that in Figure 1Bii. As it stands, it remains unclear whether Doc2B has no effect on slow burst.

4) In the Discussion section and elsewhere in the manuscript: The increase in burst amplitude in the presence of Doc2B is interpreted as promotion of a step towards priming upstream of fusion. It is not obvious why this could not as well be a consequence of Doc2B inhibiting release, which could also lead to a larger RRP and could at the same time explain the reduction of the sustained phase. While the data may contain evidence for rejecting such an alternative hypothesis it should at least be explicitly discussed in the Discussion section.

5. The authors state that chromaffin cells do not express Doc2A (Subsection “Doc2B requires synaptotagmin-1 and Munc13-2 to stimulate priming”). However, Figure 2. Shows imaging data from Doc2A/B double knockout cells. How were these cells generated and why (not found in Materials and methods section)? All the other data are apparently from Doc2B single knockouts. Is there endogenous Doc2A in mouse chromaffin cells and if so how does that impact the conclusions?

6) Figure 2 needs the quantification of CONTROL. As it stands, given the apparent Doc2B labeling in DKO cells, Figure 2 is inconclusive. The intensity values in Figure 2 include the non-Doc2B-specific fluorescence in Control, which one would need to subtract to assess expression levels of the different constructs. Expression levels of different constructs may actually vary dramatically if that background subtraction is done. Also, for comparison staining intensity of unmodified WT cells would be important to quantify to determine the overexpression factors. This relates to point 2 above.

7) In Figure 3 expression of Doc2B-EGFP appears to be much higher than that of Doc2B DN-EGFP, could this affect protein localization? In contrast Figure 2 indicates that expression of the DN mutant is similar or higher than WT.

8) What are the Doc2B expression levels for the different KO cells as used in Figure 1 – are they similar in WT, syt-1, syt-7, Munc13-2 knockout cells?

9) Figure 4 indicates that the Ca^2+^-dependent priming as indicated by relative burst amplitude change is even higher in Doc2B KO than in Doc2B overexpressing cells. How does this fit with the author's conclusion that Doc2B is the Ca^2+^ sensor for priming?

10) In the long pulse stimulations of Figure 5 there seems to be no obvious exhaustion and it is not clear why the cumulative Cm increase is equated to RRP. Are the ratios of Cm responses informative for quantification of pool size? Is there more refilling between pulses than in experiments with WT chromaffin cells? Could the negative control data be included?

11) In Figure 6: How do we know the increased labeling is due to syntaxin availability and not due to direct labeling of Doc2B? Has the labeling experiment been performed in Stx-1 KO cells +/- Doc2B? The negative control data (Doc2B KO) should be included in D-H.

12) In Figure 8 it is not clear what the effect of Doc2B ∆MID is (if any) – the control (Doc2B KO) needs to be included for this data set.

Reviewer 2:

1) My major concerns deals with the central conclusion of the manuscript implicating now Doc2B as a Ca^2+^-sensor for vesicle priming, a notion that is largely based on the observation that the DN mutant and the 6A mutant abolishes any Ca^2+^-dependency of the vesicle priming reaction (Figure 3 and Figure 4). In general, this notion seems to be difficult to reconcile with previous conclusions by the authors implicating Doc2B as the Ca^2+^-sensor for spontaneous release in neurons spontaneous (Groffen et al., 2010) and others suggesting a function as a Ca^2+^-sensor for asynchronous transmitter release (Yao et al., 2011). Furthermore, the authors themselves have previously reported, otherwise (Pinheiro et al., 2013). By analyzing the exocytotic burst size over a wide range of preflash [Ca^2+^]I in wild type and Doc2B-KO cells, they could show that Doc2B has no obligatory role in driving the Ca^2+^-dependent priming reaction upstream of the slowly releasable pool. Also in the present manuscript the authors find the Ca^2+^-dependency of the priming reaction of chromaffin vesicles to be unperturbed in cells of Doc2B null mutants when compared with controls (see Figure 3 and Figure 4).

While the lack of effect with null mutant cells could be readily explained by functionally redundant isoforms, it is similarly possible that mutating Doc2B endows special properties that allow the protein to bypass the classical priming reaction. Previous studies using similar mutations have indeed suggested that mutations in the C2A region of Doc2B (intended to disrupt Ca^2+^-binding) resulted in an anomalous increase in constitutive membrane binding and thereby provide Doc2B with novel functional properties (Gaffaney et al., 2014). Thus, I'm not convinced that these experiments with the mutant proteins (DN and 6A mutation) provide clear evidence that Doc2B acts as a Ca^2+^-sensor in vesicle priming. The apparent gain-of-function mutation may rather result from anomalous membrane binding activity. The latter is only displayed as exemplary results (Figure 3 and Figure 4) and should be presented in a quantitative fashion. I also recommend to study the dependence of the burst size on preflash [Ca^2+^]i over a wider range of Ca^2+^- concentrations, as presented for example by Pinheiro et al., 2013 (Figure 5), to provide a better understanding of the underlying mechanisms.

2) Previously, the authors have shown that overexpressing Doc2b increased significantly the total number of vesicles per cell as well as the number of membrane-proximal vesicles (Pinheiro et al., 2013). This raises the question, to what extent apparent changes in priming with the DN and 6A mutation are accompanied by alterations in vesicle number and whether a similar phenotype can be observed with the mutant proteins. In this context, I do not understand the authors' statement in subsection “Doc2B supports vesicle priming as a calcium sensor”: "The DN-mutant was expressed at similar levels as WT Doc2B, and VAMP2/synaptobrevin-2 staining also showed similar levels, consistent with an unperturbed granule density in the presence of Doc2B expression (Figure 2)." The authors should clarify this point.

3) The most prominent phenotype in Doc2B KO seems to be the enhanced sustained release under prolonged elevation of [Ca^2+^]i. This action of Doc2B is dependent on its MID-domain (Figure 8) but curiously not on its interaction with Munc 13-2 (Figure 1). The authors hypothesized a secondary function of Doc2B that blocks interactions with CAPS-proteins leading to inhibition of sustained release. While intriguing, the authors provide no evidence for an interaction between Doc2B MID domain and CAPS-protein. Furthermore, functional studies of Doc2B action in CAPS KO are required to draw such conclusions, in absence of which the readers are left with rather incomplete understanding of the inhibitory action of Doc2B. Thus, it might be helpful to study the Doc2B-CAPS interactions to better clarify the mechanism of inhibitory Doc2B function in chromaffin cells.

4) It remains unclear why specifically the RRP but not the IRP is altered with Doc2B DN mutant expression.

---

## [Author Response]

Essential revisions:Reviewer 1:1) In Figure 1 an important control needs to be included, the overexpression of Doc2B in WT chromaffin cells that are not devoid of endogenous Doc2B background. This is important for assessment of the data in the Syt-1, Syt-7 and Munc13-2 KO cells, which presumably do express WT level Doc2B.

The reviewer is right; overexpression in WT chromaffin cells is an important control. We have now performed this control and included it as Figure 1—figure supplement 1. Due to the already very large Figure 1, we could not include it there. The data shows an increase in primed vesicle pools (and a decrease in the sustained component). In contrast to Doc2B over-expression in KO cells, the SRP was also increased by Doc2B expression in WT cells (CD1 mice). We point out and discuss this finding in the revised manuscript.

2) Figure 1 legend states overexpression. How do Doc2B expression levels in Doc2B KO cells compare quantitatively with Doc2B levels in WT chromaffin cells? The authors do not convincingly address the possibility that Doc2B abundance may be responsible for inhibition of the sustained phase and the reported effects seen in Syt-1 KO and Munc13-2 KO cells (Figure 1). Syt-1 KO and Munc13-2 KO cells contain endogenous proteins and the overexpression of Doc2B WT may have introduced significantly more protein above background (this is unknown because this data is not included but pSFV is known to induce very high overexpression levels). In case of higher Doc2B protein expression in Syt1 KO+Doc2B WT versus Syt1 KO, there is a possibility that this difference may explain the observed decrease. The same could be said for the Munc13-2 KO experiment (Figure 1).

The presentation of immunostainings in Figure 2 was not clear, because the background level was not included explicitly, so the overexpression level was not clear. This has now been corrected by performing new stainings (see also below). As for the rest of the question, we realized that we did not define the rationale for the experiment: it was the purpose of our experiment to induce significantly more protein above background, to push the system to show a clear effect of Doc2B. The question is whether that level of overexpression leads to artefacts. This is not the case, because the overexpression in WT or Doc2B KO cells results in an increase of primed vesicle pools, and a decrease in sustained release. This is entirely consistent with the decrease in primed vesicle pools (seen best during repetitive depolarization stimulations) and the increase in sustained release identified in the Doc2B knockout when compared to WT littermates (Pinheiro et al., 2013). We added this rationale to the Results section.

3) Subsection “Doc2B requires synaptotagmin-1 and Munc13-2 to stimulate priming”, Figure 1Bii: The authors state that in absence of Syt-1, Doc2B has no effect on slow burst. However, the slow burst in syt1 KO is almost 50% larger than that in syt1 KO + Doc2B WT although the error bars are too large to achieve apparently p<0.05 significance. Is this due to the smaller sample size? In Figure1Ai, similar amount of difference between Doc2B KO and Doc2B KO+Doc2B WT is statistically significant, and the sample size is also larger than that in Figure 1Bii. As it stands, it remains unclear whether Doc2B has no effect on slow burst.

As mentioned in the figure legend, we measured 18 syt-1 KO cells and 20 syt-1 KO cells overexpressing Doc2B, which are reasonable numbers. However, the slow burst is the hardest phase to quantify, because it is sandwiched between the fast burst and the sustained component. Therefore, it is always a risk that the reduction in the sustained component (as seen with Doc2B overexpression) could leak into the slow burst and appear as a reduction there. Therefore, we feel we cannot make strong statements about the slow burst if the other phases are changed as well. We have now pointed this out in the text.

4) In the Discussion section and elsewhere in the manuscript: The increase in burst amplitude in the presence of Doc2B is interpreted as promotion of a step towards priming upstream of fusion. It is not obvious why this could not as well be a consequence of Doc2B inhibiting release, which could also lead to a larger RRP and could at the same time explain the reduction of the sustained phase. While the data may contain evidence for rejecting such an alternative hypothesis it should at least be explicitly discussed in the Discussion section.

This is a good point, and we now discuss it in the Discussion section. In our previous work (Pinheiro et al., 2013), we could not make this distinction, but now we can, because we here identify several mutations, where the increase in RRP is absent, but the decrease in sustained release is still detected. So, we have separated those two functions by mutation (see first paragraph of Discussion section).

5. The authors state that chromaffin cells do not express Doc2A (Subsection “Doc2B requires synaptotagmin-1 and Munc13-2 to stimulate priming”). However, Figure 2. Shows imaging data from Doc2A/B double knockout cells. How were these cells generated and why (not found in Materials and methods section)? All the other data are apparently from Doc2B single knockouts. Is there endogenous Doc2A in mouse chromaffin cells and if so how does that impact the conclusions?

Doc2A is not expressed in chromaffin cells (Friedrich et al., 2008; Pinheiro et al., 2013), which we mentioned in the manuscript. The staining was done in Doc2A/B double-KO cells, because the Doc2B single KO mouse line was not available in the laboratory when these experiments were performed. This should not have any impact on results, because Doc2A is not expressed. We have now added this explanation and the description of this mouse line to the Materials and methods section.

6) Figure 2 needs the quantification of CONTROL. As it stands, given the apparent Doc2B labeling in DKO cells, Figure 2 is inconclusive. The intensity values in Figure 2 include the non-Doc2B-specific fluorescence in Control, which one would need to subtract to assess expression levels of the different constructs. Expression levels of different constructs may actually vary dramatically if that background subtraction is done. Also, for comparison staining intensity of unmodified WT cells would be important to quantify to determine the overexpression factors. This relates to point 2 above.

The reviewer is correct. We have now replaced Figure 2 with new data, which includes stainings of controls. Staining of WT cells (cells from Doc2B WT mice) in parallel show that our antibody cannot detect endogenous protein in immunocytochemistry and produces some non-specific staining, probably due the presence of other epitopes in DKO cells. Nevertheless, expression of Doc2B in mouse adrenal chromaffin cells has been clearly shown by in situ hybridisation (Friedrich et al., 2006 – supplementary materials), and confirmed by analysis of the knockout cells, incl. the ability to rescue the phenotype with Doc2B expression (Pinheiro et al., 2013). Thus, the fact that the antibody cannot detect the endogenous protein is most likely an antibody sensitivity problem. This does not affect its ability to detect overexpressed protein, as it is present at higher levels.

7) In Figure 3 expression of Doc2B-EGFP appears to be much higher than that of Doc2B DN-EGFP, could this affect protein localization? In contrast Figure 2 indicates that expression of the DN mutant is similar or higher than WT.

We tested this possibility, but observed that the relative expression levels of the Doc2B DN and WT protein cannot be assessed by comparing their EGFP-fluorescence in live cells, because the two constructs do not have the same localization. In our Doc2B WT construct, when stimulating with high-K (so that the protein goes to the membrane), the total fluorescence is decreased – this is reversible. This is probably due to self-quenching of EGFP when accumulating at the membrane, or recruitment to parts of the plasma membrane outside of the confocal slice. At the suggestion of reviewer #2 we have now quantified live cell imaging experiments in a number of cells and include the quantification in a separate figure (new Figure 3). Since absolute intensities are not informative, quantifications are done on ratios (total fluorescence / fluorescence inside the cell). In addition, please note that the fusion constructs are not used for staining or electrophysiological experiments (for those experiments bicistronic constructs were used), therefore they cannot be directly compared to Figure 2.

8) What are the Doc2B expression levels for the different KO cells as used in Figure 1 – are they similar in WT, syt-1, syt-7, Munc13-2 knockout cells?

We have performed Western Blot against Doc2B of Syt-1 KO and Syt-7 KO cells and included it in Figure 1—figure supplement 2. The endogenous protein is hard to detect with the antibody, but no overt changes were noted. Our Munc13-2 KO mouse colony is currently not running, and we have not obtained any glands. Please note – this is a point related to the point 2 above – that no conclusion in our manuscript requires the Doc2B expression levels to be exactly the same in the KOs. The point of our experiment in Figure 1 was to use overexpression to push the effect of Doc2B to the maximum, and test which of the other secretory proteins it depends on. We have added this rationale in the Results section.

9) Figure 4 indicates that the Ca^2+^-dependent priming as indicated by relative burst amplitude change is even higher in Doc2B KO than in Doc2B overexpressing cells. How does this fit with the author's conclusion that Doc2B is the Ca^2+^ sensor for priming?

This is already discussed in the Discussion section: we show here that Doc2B is a Ca^2+^-sensor for priming, however there must be an overlapping or redundant pathway in chromaffin cells, since priming is still present in the KOs. We do not think this is very surprising. Munc13-2 has several Ca^2+^-binding sites and can probably be activated without Doc2B.

10) In the long pulse stimulations of Figure 5 there seems to be no obvious exhaustion and it is not clear why the cumulative Cm increase is equated to RRP. Are the ratios of Cm responses informative for quantification of pool size? Is there more refilling between pulses than in experiments with WT chromaffin cells? Could the negative control data be included?

The release until the end was found to be similar to the RRP in flash experiments, therefore we call it “RRP”, but we discuss that it might not be exactly the same pool as the RRP found in flash data. We don’t find the ratios very informative, since the depression is very small (or absent) in many cells and using ratios would therefore lead us to leave out many cells, making the analysis unrepresentative. There are no ‘negative control data’ here (or we don’t understand what the reviewer means): as we describe, expression of Doc2B WT, or Doc2B DN led to strong inhibition of Calcium-currents; therefore, a comparison to uninfected cells is not meaningful.

11) In Figure 6: How do we know the increased labeling is due to syntaxin availability and not due to direct labeling of Doc2B? Has the labeling experiment been performed in Stx-1 KO cells +/- Doc2B? The negative control data (Doc2B KO) should be included in D-H.

The antibody used is specific for syntaxin-1, as we have shown in another paper using BoNT/C expressing cells (Toft-Bertelsen et al., 2016). We now point this out in the manuscript. In that paper, we also showed that staining is increased within minutes when dispersing syntaxin-clusters directly using cholesterol extraction. In the present manuscript we show that syntaxin-1 staining is not increased when we mutate Doc2B to not bind to SNAREs (KE-mutant), which is another confirmation of the specificity of the effect (and the antibody). We do not have access to syntaxin-1 KO mice. We have included quantification of the KO condition in panels D-H.

12) In Figure 8 it is not clear what the effect of Doc2B ∆MID is (if any) – the control (Doc2B KO) needs to be included for this data set.

We have now included the Doc2B KO condition and discussed the effect of ∆MID clearer in the revised manuscript.

Reviewer 2:1) My major concerns deals with the central conclusion of the manuscript implicating now Doc2B as a Ca^2+^-sensor for vesicle priming, a notion that is largely based on the observation that the DN mutant and the 6A mutant abolishes any Ca^2+^-dependency of the vesicle priming reaction (Figure 3 and Figure 4). In general, this notion seems to be difficult to reconcile with previous conclusions by the authors implicating Doc2B as the Ca^2+^-sensor for spontaneous release in neurons spontaneous (Groffen et al., 2010) and others suggesting a function as a Ca^2+^-sensor for asynchronous transmitter release (Yao et al., 2011). Furthermore, the authors themselves have previously reported, otherwise (Pinheiro et al., 2013). By analyzing the exocytotic burst size over a wide range of preflash [Ca^2+^]I in wild type and Doc2B-KO cells, they could show that Doc2B has no obligatory role in driving the Ca^2+^-dependent priming reaction upstream of the slowly releasable pool. Also in the present manuscript the authors find the Ca^2+^-dependency of the priming reaction of chromaffin vesicles to be unperturbed in cells of Doc2B null mutants when compared with controls (see Figure 3 and Figure 4).While the lack of effect with null mutant cells could be readily explained by functionally redundant isoforms, it is similarly possible that mutating Doc2B endows special properties that allow the protein to bypass the classical priming reaction. Previous studies using similar mutations have indeed suggested that mutations in the C2A region of Doc2B (intended to disrupt Ca^2+^-binding) resulted in an anomalous increase in constitutive membrane binding and thereby provide Doc2B with novel functional properties (Gaffaney et al., 2014). Thus, I'm not convinced that these experiments with the mutant proteins (DN and 6A mutation) provide clear evidence that Doc2B acts as a Ca^2+^-sensor in vesicle priming. The apparent gain-of-function mutation may rather result from anomalous membrane binding activity. The latter is only displayed as exemplary results (Figure 3 and Figure 4) and should be presented in a quantitative fashion. I also recommend to study the dependence of the burst size on preflash [Ca^2+^]i over a wider range of Ca^2+^- concentrations, as presented for example by Pinheiro et al., 2013 (Figure 5), to provide a better understanding of the underlying mechanisms.

We have now performed new experiments and included a quantification of the membrane binding activity of Doc2B WT, DN, 6A, and ΔMID in a new figure (Figure 3). This shows – as we had previously described qualitatively – that WT and ΔMID cycle on and off the plasma membrane upon treatment with High-K^+^ solution. Also the DN cycles to a minor degree (as we had mentioned also before), but it also displays a much higher membrane localization at rest than the WT; thus it is only a minor fraction of the DN mutant that cycles, since most resides at the membrane. The 6A-mutation does not cycle on and off the membrane, but remains at the plasma membrane. The reviewer raises the question whether the DN mutation has ‘anomalous increase in constitutive membrane binding and thereby provides Doc2B with novel functional properties’. The decisive experiments that we performed here was to study the effect of expressing Doc2B WT and DN when stimulated from a low and a high basal-[Ca^2+^]. The striking result is that the DN is a gain-of-function mutant when studied from a low basal-[Ca^2+^], but – when studied from a high basal [Ca^2+^] – it is indistinguishable from the WT. This latter fact indicates that the DN is not special or anomalous, except that it is already abundant at the membrane at very low [Ca^2+^]. The WT protein would not be at the membrane at low basal-[Ca^2+^], and thus the gain-of-function of the DN-mutant is entirely understandable from the trafficking behavior. The only exception is the fusion time constant of the RRP, which is significantly faster for the DN-mutant when studied from high basal-[Ca^2+^] (we commented on this also in the first version of the manuscript). In addition, we see basically the same behavior by another mutant – the 6A-mutant – which is unable to bind to Ca^2+^ at all, but which is also constitutively at the membrane. Also, this mutation is gain-of-function when studied from low basal-[Ca^2+^], although it is slightly inferior to the WT at high basal [Ca^2+^]. This permanent membrane localization of the 6A-mutation sheds new light on this mutation, which was previously used to argue that Doc2B might have Ca^2+^-independent effect (Pang et al., 2011); now we see that also this mutation bypasses Ca^2+^-binding by being permanently at the plasma membrane. We have not been able to extend the range of preflash [Ca^2+^] that we study, because of the huge amount of work that would be involved, in combination with the modest yield of cells from a single Doc2B embryo. As far as we can see, the current data provide very strong evidence for our conclusion that Ca^2+^-binding to Doc2B drives it to the membrane, which leads to membrane priming.

Our data here are consistent with our previous data (Pinheiro et al., 2013), since we noted – and discussed carefully in the Discussion that calcium-dependent priming is still present in Doc2B KO cells, probably because of redundant processes/proteins (as the reviewer mentions), which can be overridden by overexpression of Doc2B DN. When comparing to data obtained in synapses, it is worth remembering that the vesicles we study here (large dense-core vesicles) are different from synaptic vesicles, and there is nothing surprising in having different functions of the protein in the two (very) different systems. Also, the groups studying synaptic vesicles cannot agree as to the function of Doc2B in asynchronous release. Nevertheless, we think the data might be reconciled, since priming in chromaffin cells is driven by SNARE-complex assembly, and Ca^2+^-dependent (due to the trafficking of Doc2B) SNARE-complex formation could very well in other systems lead to spontaneous, or even (if fast enough, which is unclear) asynchronous release.

2) Previously, the authors have shown that overexpressing Doc2B increased significantly the total number of vesicles per cell as well as the number of membrane-proximal vesicles (Pinheiro et al., 2013). This raises the question, to what extent apparent changes in priming with the DN and 6A mutation are accompanied by alterations in vesicle number and whether a similar phenotype can be observed with the mutant proteins. In this context, I do not understand the authors' statement in subsection “Doc2B supports vesicle priming as a calcium sensor”: "The DN-mutant was expressed at similar levels as WT Doc2B, and VAMP2/synaptobrevin-2 staining also showed similar levels, consistent with an unperturbed granule density in the presence of Doc2B expression (Figure 2)." The authors should clarify this point.

We have repeated the stainings, as explained above. In the process of doing this, we realized that quantification of Doc2 and Syb2 cannot be done accurately on samples stained for both proteins, because the expression and staining of Doc2 reduces the staining for Syb2, probably because of antibody interference. We have therefore performed separate stainings for Doc2 and syb2, in an attempt to solve this question. However, this didn’t change the conclusion. We can only speculate why there is a difference between the EM and the immunostaining. These are very different measures, and there are plenty of reasons these measures could turn out different. In the EM the levels of proteins were not quantified; In Immunostaining we did not quantify granules, only overall syb2-intensity. Thus, formally one possibility is that the total syb2 content is distributed between more granules after overexpression, although because these measurements were done several years apart by different people, we would not be comfortable making such a conclusion at the present time. We now pointed out this possible discrepancy in the Discussion section.

3) The most prominent phenotype in Doc2B KO seems to be the enhanced sustained release under prolonged elevation of [Ca^2+^]i. This action of Doc2B is dependent on its MID-domain (Figure 8) but curiously not on its interaction with Munc 13-2 (Figure 1). The authors hypothesized a secondary function of Doc2B that blocks interactions with CAPS-proteins leading to inhibition of sustained release. While intriguing, the authors provide no evidence for an interaction between Doc2B MID domain and CAPS-protein. Furthermore, functional studies of Doc2B action in CAPS KO are required to draw such conclusions, in absence of which the readers are left with rather incomplete understanding of the inhibitory action of Doc2B. Thus, it might be helpful to study the Doc2B-CAPS interactions to better clarify the mechanism of inhibitory Doc2B function in chromaffin cells.

We agree, studying the Doc2B-CAPS interaction directly would certainly be helpful. However, repeating the experiments in CAPS DKO cells has not been possible, because we currently don’t have CAPS DKO mice available in the laboratory. We have tried to make it clearer in our text that we are discussing the possibilities; we are not drawing a conclusion.

4) It remains unclear why specifically the RRP but not the IRP is altered with Doc2B DN mutant expression.

The IRP is larger on average, but it is far from significant. We now discuss this. The point is that the IRP is a subpool of the RRP, consisting of RRP-vesicles that are close to Ca^2+^-channels. Therefore, there are more factors involved in making an IRP-vesicle than a RRP-vesicle. Most likely another factor (number of Ca^2+^-channels?) is limiting for the IRP-vesicles.